# Adversarially Robust Models may not Transfer Better: Sufficient Conditions for Domain Transferability from the View of Regularization

## Abstract

Machine learning (ML) robustness and generalization are fundamentally correlated: they essentially concern about data distribution shift under adversarial and natural settings, respectively. Thus, it is critical to uncover their underlying connections to tackle one based on the other. On one hand, recent studies show that more robust (adversarially trained) models are more generalizable to other domains. On the other hand, there lacks of theoretical understanding of such phenomenon, and it is not clear whether there are counterexamples. In this paper, we aim to provide sufficient conditions for this phenomenon considering different factors that could affect both, such as norm of the last layer, Jacobian norm, and data augmentations (DA). In particular, we propose a general theoretical framework indicating factors that can be reformed as a function class regularization process, which could lead to improvements of domain generalization. Our analysis, for the first time, shows that "robustness" is actually not the causation for domain generalization; rather, robustness induced by adversarial training is a by-product of such function class regularization. We then discuss in details about different properties of DA and we prove that under certain conditions, DA can be viewed as regularization and therefore improve generalization. We conduct extensive experiments to verify our theoretical findings and show several counterexamples where robustness and generalization are negatively correlated when the sufficient conditions are not satisfied.

## 1 Introduction

Domain generalization (or transferability) is the task of training machine learning models with data from one or more *source* domains that can be adapted to a *target* domain, often via low-cost fine-tuning. Thus, domain generalization refers to approaches designed to address the underline{natural data distribution shift} problem (Muandet et al., 2013; Rosenfeld et al., 2021). A wide array of approaches have been proposed to address domain transferability, including fine-tuning the last layer of DNNs (Huang et al., 2018), invariant feature optimization (Muandet et al., 2013), efficient model selection for fine-tuning (You et al., 2019), and optimal transport based domain adaptation (Courty et al., 2016). Improving domain generalization has emerged as an important task in the machine learning community: for instance, it is among the key technologies to enable an autonomous driving vehicle trained in city scenarios to make correct decisions in the countryside as well.

On the other hand, robust machine learning aims to tackle the problem of underline{adversarial data distribution shift}. Both empirical and certified robust learning approaches have been proposed, such as empirical adversarial training (Madry et al., 2018) and certified defenses based on deterministic and probabilistic approaches (Cohen et al., 2019).

As domain transferability and robust machine learning tackle different kinds of data distribution shifts, this work seeks to uncover their underlying connections and tradeoffs. For instance, recent studies suggest that adversarially robust models are more domain transferable (Salman et al., 2020), which, in turn, provides new insights on improving domain generalization. However, a theoretical analysis of their relationship is still lacking, and it is unclear whether such positive correlations al-

Figure 1: Illustration of robustness and domain transferability in different conditions.

ways hold. In this paper, we take the first steps towards formally analyzing the relationship between model robustness and domain transferability to answer the following questions: *What are sufficient conditions for domain transferability? Is model robustness the cause of domain transferability? Can robustness and domain transferability be negatively correlated?*

To answer the above questions and uncover the underlying relationship between robustness and domain transferability, we propose a general theoretical framework that characterizes sufficient conditions for domain transferability from the view of function class regularization. Our analysis shows that if the function class of feature extractors is more regularized, the model based on a feature extractor trained from the function class, composed with a fine-tuned last layer, can be more transferable. Formally, we prove that there is a monotone relation between the regularization strength and a tight upper bound on the relative domain transfer loss.

Under the proposed framework, we analyze several common factors for model training, including the Jacobian norm, the last layer norm, data augmentation, and adversarial training as shown in Fig. 1. In particular, controlling the Jacobian norm and last layer norm can be viewed as function-class regularization, thus can be analyzed in our framework. We also analyze how other common regularization operations can be mapped to function class regularization. For instance, we consider noise-dependent and independent data augmentation procedures based on feature average and loss average aggregation algorithms.

We conduct extensive experiments on ImageNet (CIFAR-10 as target domain) and CIFAR-10 (SVHN as target domain) based on different models to verify our analysis. We show that regularization can control domain transferability, and robustness and domain transferability can be negatively correlated, which are counter-examples against Salman et al. (2020). Taken together, this indicates that robustness is not a cause of transferability.

**Technical Contributions.** We aim to uncover the underlying relationship between robustness and domain transferability and lay out the sufficient conditions for transferability from the view of regularization. We make both theoretical and empirical contributions.

- We propose a theoretical framework to analyze the sufficient conditions for domain transferability from the view of function class regularization. We provably show that stronger regularization on the feature extractor implies a decreased tight upper bound on the relative transferability loss; while model robustness could be arbitrary.

- We prove the tightness of our transferability upper bound, and provide the generalization bound of the relative transferability loss from the view of regularization.

- We analyze several factors such as different data augmentations (*e.g.*, rotation and Gaussian) under the framework, and show how they can be mapped to function class regularization and therefore affect transferability.

- We conduct extensive experiments on different datasets and model architectures to verify our theoretical claims. We also show several counterexamples that indicate significant negative correlation between robustness and the relative domain transferability.

## 2 RELATED WORK

**Domain Transferability** has been analyzed in different settings. Muandet et al. present a generalization bound for classification task based on the properties of the assumed prior over training environments. Rosenfeld et al. model domain transferability/generalization as an online game and show that generalizing beyond the convex hull of training environments is NP-hard, and Zhang et al. provides a generalization bound for distributions with sufficiently small H-divergence. Given the

complexity of domain transferability analysis, recent empirical studies show that adversarially robust models transfer better (Salman et al., 2020). In this paper, we aim to relax the assumptions and focus on understanding the domain transferability from the view of regularization and theoretically show whether "robustness" is indeed a causation for transferability or not.

**Model Robustness** is an important topic given recent diverse adversarial attacks (Goodfellow et al., 2014; Carlini & Wagner, 2017). These attacks may be launched without access to model parameters (Tu et al., 2019) or even with the model predicted label alone (Chen et al., 2020). Different approaches have been proposed to improve model robustness against adversarial attack. Adversarial training has been shown to be effective empirically (Madry et al., 2018; Zhang et al., 2019a; Miyato et al., 2018). Some studies have shown that robustness is property related to other model characteristics, such as transferability and invertibility (Engstrom et al., 2019).

## 3 SUFFICIENT CONDITIONS FOR DOMAIN TRANSFERABILITY

In this section, we theoretically analyze the problem of domain transferability from the view of regularization and discuss some sufficient conditions for good transferability. All of the proofs are provided in Section A in the appendix.

**Notations.** We denote the input space as $\mathcal{X}$; the feature space as $\mathcal{Z}$ and the output space as $\mathcal{Y}$. Let the fine-tuning function class be $g \in \mathcal{G}$. Given a feature extractor $f : \mathcal{X} \to \mathcal{Z}$ and a fine-tuning function $g : \mathcal{Z} \to \mathcal{Y}$, the full model is $g \circ f : \mathcal{X} \to \mathcal{Y}$. We denote $\mathcal{P}_{\mathcal{X} \times \mathcal{Y}}$ as the set of distributions on $\mathcal{X} \times \mathcal{Y}$. The loss function is denoted by $\ell : \mathcal{Y} \times \mathcal{Y} \to \mathbb{R}_+$, and the population loss based on data distribution $\mathcal{D} \in \mathcal{P}_{\mathcal{X} \times \mathcal{Y}}$ and a model $g \circ f$ is defined as

$$\ell_{\mathcal{D}}(g \circ f) := \mathbb{E}_{(x,y) \sim \mathcal{D}}[\ell(g \circ f(x), y)].$$

Before diving into the details, we first provide the following example to illustrate why one might investigate domain transferability from the view of regularization.

### 3.1 EXAMPLE: ROBUSTNESS AND TRANSFERABILITY ARE INDEPENDENT

In this subsection, we construct a simple example where domain transferability depends on regularization, yet domain transferability and robustness are *independent*. Moreover, this example serves as motivation to consider domain transferability from the view of regularization.

Given the source and target distributions $\mathcal{D}_S, \mathcal{D}_T \in \mathbb{P}_{\mathcal{X} \times \mathcal{Y}}$, we denote their marginal distributions on the input space $\mathcal{X}$ as $\mathcal{D}_S^{\mathcal{X}}$ and $\mathcal{D}_T^{\mathcal{X}}$, respectively. We consider the case that $\mathcal{X} \subset \mathbb{R}^m$ being a low-dimensional manifold in $\mathbb{R}^m$, and $\mathcal{Y} = \mathbb{R}^d$. Given an input $x \in \mathcal{X}$, the ground truth target for the source domain is $y_S(x)$ generated by a function $y_S : \mathbb{R}^m \to \mathbb{R}^d$. Similarly, we define $y_T$ for the target domain. In this example, for simplicity, we neglect the fine-tuning process but directly consider learning a function $f : \mathbb{R}^m \to \mathbb{R}^d$ with a norm $\|\cdot\|$ on $\mathbb{R}^d$. For the source domain we have the population loss:

$$\ell_{\mathcal{D}_S}(f) = \mathbb{E}_{x \sim \mathcal{D}_S^{\mathcal{X}}}[\|f(x) - y_S(x)\|].$$

A distribution $\mathcal{D} \in \mathbb{P}_{\mathcal{X}}$ on the input space $\mathcal{X}$, defines a norm of a function $f : \mathbb{R}^m \to \mathbb{R}^d$ as

$$\|f\|_{\mathcal{D}} := \mathbb{E}_{x \sim \mathcal{D}}[\|f(x)\|],$$

where we view two functions $f_1, f_2$ as the same if $\|f_1 - f_2\|_{\mathcal{D}} = 0$. Therefore, we can define the source domain loss $\ell_{\mathcal{D}_S}(f) = \|f - y_S\|_{\mathcal{D}_S^{\mathcal{X}}}$ and the target domain loss $\ell_{\mathcal{D}_T}(f) = \|f - y_T\|_{\mathcal{D}_T^{\mathcal{X}}}$.

For the sake of illustration, we consider the simple case where the input distributions $\mathcal{D}_S^{\mathcal{X}}, \mathcal{D}_T^{\mathcal{X}}$ are the same, and hence we denote $\mathcal{D} = \mathcal{D}_S^{\mathcal{X}} = \mathcal{D}_T^{\mathcal{X}}$. Note that $y_S$ and $y_T$ are different.

Denoting a function space $\mathcal{F} = \{f : \mathbb{R}^m \to \mathbb{R}^d \mid \|f\|_{\mathcal{D}} < \infty\}$, we assume that $y_S, y_T \in \mathcal{F}$ and we can compare $f, y_S, y_T$ in the same space. Therefore, given $c > 0$ as a regularization parameter, the domain transferability problem can be defined as:

$$\text{Learning a source model:} \qquad f_c^{\mathcal{D}_S} \in \arg\min_{f \in \mathcal{F}} \ell_{\mathcal{D}_S}(f), \quad \text{s.t. } \|f\|_{\mathcal{D}} \leq c; \qquad (1)$$

$$\text{Testing on a target domain:} \qquad \ell_{\mathcal{D}_T}(f_c^{\mathcal{D}_S}),$$

where the minimizer $f_c^{\mathcal{D}_S} := y_S \min\{1, \frac{c}{\|y_S\|_{\mathcal{D}}}\}$, the source domain loss is $\ell_{\mathcal{D}_S}(f) = \|f - y_S\|_{\mathcal{D}}$, and the target domain loss is $\ell_{\mathcal{D}_T}(f) = \|f - y_T\|_{\mathcal{D}}$. We prove in Proposition 3.1 that $f_c^{\mathcal{D}_S}$ is indeed a minimizer of (1) and provide an intuitive illustration in Figure 2.

We show that the robustness can be independent to domain transferability as follows. Consider the adversarial robustness of $f_c^{\mathcal{D}_S}$ on an input $x \in \mathcal{X}$ (e.g., $\max_{\delta:\|\delta\|_2 \le \epsilon} \ell(f_c^{\mathcal{D}_S}(x + \delta), y_S(x)))$. Since the transferred loss $\ell_{\mathcal{D}_T}(f_c^{\mathcal{D}_S})$ only evaluates $f_c^{\mathcal{D}_S}$ on $\mathcal{X}$ which is a low-dimensional manifold in $\mathbb{R}^m$, an adversarial perturbation $\delta \in \mathbb{R}^m$ could make $x + \delta \in \mathbb{R}^m \backslash \mathcal{X}$ when the loss function is sufficiently big outside the manifold $\mathcal{X}$. Therefore, the robustness could be arbitrarily bad without changing the value of $\ell_{\mathcal{D}_T}(f_c^{\mathcal{D}_S})$, i.e., the performance of the source model on the target domain.

As we can see, *the robustness is independent to domain transferability in this example*. On the contrary, if we change the perspective to consider the the regularization parameter $c$, we have the following interesting finding. An illustration of the finding is shown in Figure 2, and a more formal statement is provided in Proposition 3.1.

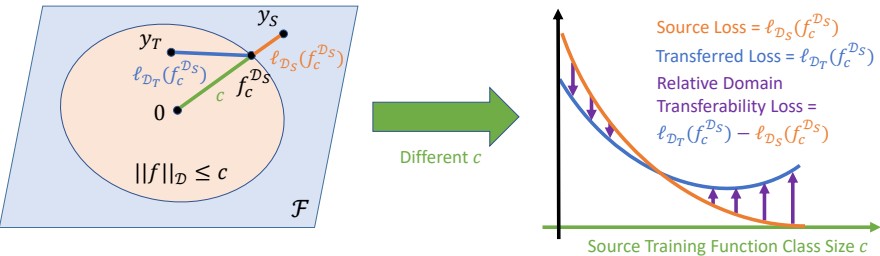

Figure 2: The left figure illustrates the example in the function space $\mathcal{F}$ given a regularization parameter $c$. The right figure shows the relations between domain transferability and the $c$. In this example, the weaker the regularization effect (greater $c$) is, the greater the relative domain transferability loss (violet arrow becomes).

**Proposition 3.1.** *Given the problem defined above, $f_c^{\mathcal{D}_S}$ is a minimizer of equation 1. If $c \ge c' \ge 0$, then the relative domain transferability loss $\ell_{\mathcal{D}_T}(f_c^{\mathcal{D}_S}) - \ell_{\mathcal{D}_S}(f_c^{\mathcal{D}_S}) \ge \ell_{\mathcal{D}_T}(f_{c'}^{\mathcal{D}_S}) - \ell_{\mathcal{D}_S}(f_{c'}^{\mathcal{D}_S})$.*

We can see that robustness is not sufficient to characterize domain transferability. However, there is a monotone relation between the regularization strength and the relative transferability loss, where adversarial robustness could be arbitrary. Similar behavior is also observed in our experiments, as we will discuss in Section 4. Naturally, these findings motivate the study of the connections between the regularization of the training process and domain transferability in general, as we consider next.

## 3.2 UPPER BOUND OF THE RELATIVE DOMAIN TRANSFERABILITY

In this subsection, we consider the general transferability problem with fine-tuning. We prove that there is a monotone relationship between the regularization strength and relative domain transferability loss. We also present a tight upper bound on the relative domain transferability loss. Denote the training algorithm as $A$ that takes a data distribution $\mathcal{D}$ and outputs a feature extractor $f_A^{\mathcal{D}} \in \mathcal{F}_A$ chosen from a function class $\mathcal{F}_A$ and a fine-tuning function $g_A^{\mathcal{D}} \in \mathcal{G}$. Next we formally define relative domain transferability.

**Definition 1** (Relative Domain Transferability Loss). *Given the training algorithm $A$ and a pair of distributions $\mathcal{D}_S, \mathcal{D}_T \in \mathcal{P}_{\mathcal{X} \times \mathcal{Y}}$, the relative domain transferability loss between $\mathcal{D}_S, \mathcal{D}_T$ is defined to be the difference of fine-tuned losses, i.e.,*

$$\tau(A; \mathcal{D}_S, \mathcal{D}_T) := \inf_{g \in \mathcal{G}} \ell_{\mathcal{D}_T}(g \circ f_A^{\mathcal{D}_S}) - \ell_{\mathcal{D}_S}(g_A^{\mathcal{D}_S} \circ f_A^{\mathcal{D}_S}).$$

Note that the training algorithm $A$ is not required to be optimal, i.e., it could be the case that $\ell_{\mathcal{D}_S}(g_A^{\mathcal{D}_S} \circ f_A^{\mathcal{D}_S}) > \inf_{g \in \mathcal{G}, f \in \mathcal{F}_A} \ell_{\mathcal{D}_S}(g \circ f)$. As we can see, the smaller $\tau(A; \mathcal{D}_S, \mathcal{D}_T)$ is, the better the model's relative performance becomes on the target domain.

Another perspective of Definition 1 is that $\inf_{g \in \mathcal{G}} \ell_{\mathcal{D}_T}(g \circ f_A^{\mathcal{D}_S}) = \ell_{\mathcal{D}_S}(g_A^{\mathcal{D}_S} \circ f_A^{\mathcal{D}_S}) + \tau(A; \mathcal{D}_S, \mathcal{D}_T)$. From this perspective, the transferred loss is the source loss plus an additional term to be upper bounded by a certain distance metric between the source and target distributions – as is common in the literature of domain adaptation (e.g., Ben-David et al. (2007); Zhao et al. (2019)). The key question of the "distance metric" remains unanswered. To this end, we propose the following.

**Definition 2** (($\mathcal{G}, \mathcal{F}$)-pseudometric). *Given a fine-tuning function class $\mathcal{G}$, a feature extractor function class $\mathcal{F}$ and distributions $\mathcal{D}_S, \mathcal{D}_T \in \mathcal{P}_{\mathcal{X} \times \mathcal{Y}}$, the ($\mathcal{G}, \mathcal{F}$)-pseudometric between $\mathcal{D}_S, \mathcal{D}_T$ is*

$$d_{\mathcal{G},\mathcal{F}}(\mathcal{D}_S, \mathcal{D}_T) := \sup_{f \in \mathcal{F}} |\inf_{g \in \mathcal{G}} \ell_{\mathcal{D}_S}(g \circ f) - \inf_{g \in \mathcal{G}} \ell_{\mathcal{D}_T}(g \circ f)|.$$

*Since the fine-tuning function class is usually simple and fixed, we will use $d_{\mathcal{F}}$ as an abbreviation in the context where $\mathcal{G}$ is clear.*

It can be easily verified that $d_{\mathcal{G},\mathcal{F}}$ is a pseudometric that measures the *distance* between two distributions, as shown in the following proposition.

**Proposition 3.2.** $d_{\mathcal{G},\mathcal{F}}(\cdot, \cdot) : \mathcal{P}_{\mathcal{X} \times \mathcal{Y}} \times \mathcal{P}_{\mathcal{X} \times \mathcal{Y}} \to \mathbb{R}_+$ *satisfies the following properties; (Symmetry) $d_{\mathcal{G},\mathcal{F}}(\mathcal{D}_S, \mathcal{D}_T) = d_{\mathcal{G},\mathcal{F}}(\mathcal{D}_T, \mathcal{D}_S)$; (Triangle Inequality) $\forall \mathcal{D}' \in \mathcal{P}_{\mathcal{X} \times \mathcal{Y}} : d_{\mathcal{G},\mathcal{F}}(\mathcal{D}_S, \mathcal{D}_T) \leq d_{\mathcal{G},\mathcal{F}}(\mathcal{D}_S, \mathcal{D}') + d_{\mathcal{G},\mathcal{F}}(\mathcal{D}', \mathcal{D}_T)$; (Weak Zero Property) $\forall \mathcal{D} \in \mathcal{P}_{\mathcal{X} \times \mathcal{Y}} : d_{\mathcal{G},\mathcal{F}}(\mathcal{D}, \mathcal{D}) = 0$.*

In this section, we consider a fixed fine-tuning function class $\mathcal{G}$ and feature extractor function class $\mathcal{F}_A$ given by the training algorithm $A$. Thus, we denote $d_{\mathcal{G},\mathcal{F}}$ as $d_{\mathcal{F}_A}$ for the remainder of the paper. With the definition of $d_{\mathcal{F}_A}$, we can derive the following result.

**Theorem 3.1.** *Given a training algorithm $A$, for $\forall \mathcal{D}_S, \mathcal{D}_T \in \mathcal{P}_{\mathcal{X} \times \mathcal{Y}}$ we have*

$$\tau(A; \mathcal{D}_S, \mathcal{D}_T) \leq d_{\mathcal{F}_A}(\mathcal{D}_S, \mathcal{D}_T),$$

*or equivalently,* $\quad \inf_{g \in \mathcal{G}} \ell_{\mathcal{D}_T}(g \circ f_A^{\mathcal{D}_S}) \leq \ell_{\mathcal{D}_S}(g_A^{\mathcal{D}_S} \circ f_A^{\mathcal{D}_S}) + d_{\mathcal{F}_A}(\mathcal{D}_S, \mathcal{D}_T).$

**Interpretation:** As we can see, the above theorem provides sufficient conditions for good domain transferability. There is a monotone relation between the regularization strength and $d_{\mathcal{F}_A}(\mathcal{D}_S, \mathcal{D}_T)$, i.e., the upper bound on the relative domain transferability loss $\tau(A; \mathcal{D}_S, \mathcal{D}_T)$. More explicitly, if a training algorithm $A'$ has $\mathcal{F}_{A'} \subseteq \mathcal{F}_A$, then $d_{\mathcal{F}_{A'}}(\mathcal{D}_S, \mathcal{D}_T) \leq d_{\mathcal{F}_A}(\mathcal{D}_S, \mathcal{D}_T)$. Moreover, small $d_{\mathcal{F}_A}(\mathcal{D}_S, \mathcal{D}_T)$ implies good relative domain transferability. From this perspective, we can see that we need both small $d_{\mathcal{F}_A}(\mathcal{D}_S, \mathcal{D}_T)$ and small source loss $\ell_{\mathcal{D}_S}(g_A^{\mathcal{D}_S} \circ f_A^{\mathcal{D}_S})$ to guarantee good absolute domain transferability. Note that there is a possible trade-off, i.e., with $\mathcal{F}_A$ being smaller, $d_{\mathcal{F}_A}(\mathcal{D}_S, \mathcal{D}_T)$ decreases but possibly $\ell_{\mathcal{D}_S}(g_A^{\mathcal{D}_S} \circ f_A^{\mathcal{D}_S})$ increases due to the limited power of $\mathcal{F}_A$. On the other hand, there may not be such trade-off if $\mathcal{D}_S$ and $\mathcal{D}_T$ are close enough such that $d_{\mathcal{F}_A}(\mathcal{D}_S, \mathcal{D}_T)$ is small.

To make the upper bound more meaningful, we need to study the tightness of it.

**Theorem 3.2.** *Given any source distribution $\mathcal{D}_S \in \mathcal{P}_{\mathcal{X} \times \mathbb{R}^d}$, any fine-tuning function class $\mathcal{G}$ where $\mathcal{G}$ includes the zero function, and any training algorithm $A$, denote*

$$\epsilon := \ell_{\mathcal{D}_S}(g_A^{\mathcal{D}_S} \circ f_A^{\mathcal{D}_S}) - \inf_{g \in \mathcal{G}, f \in \mathcal{F}_A} \ell_{\mathcal{D}_S}(g \circ f).$$

*We assume some properties of the loss function $\ell : \mathbb{R}^d \times \mathbb{R}^d \to \mathbb{R}_+$: it is differentiable and strictly convex w.r.t. its first argument; $\ell(y, y) = 0$ for any $y \in \mathbb{R}^d$; and $\lim_{r \to \infty} \inf_{y : \|y\|_2 = r} \ell(\vec{0}, y) = \infty$, where $\vec{0}$ is the zero vector. Then, for any distribution $\mathcal{D}^{\mathcal{X}}$ on $\mathcal{X}$, there exist some distributions $\mathcal{D}_T \in \mathcal{P}_{\mathcal{X} \times \mathbb{R}^d}$ with its marginal on $\mathcal{X}$ being $\mathcal{D}^{\mathcal{X}}$ such that*

$$\tau(A; \mathcal{D}_S, \mathcal{D}_T) \leq d_{\mathcal{F}_A}(\mathcal{D}_S, \mathcal{D}_T) \leq \tau(A; \mathcal{D}_S, \mathcal{D}_T) + \epsilon,$$

*or equivalently $d_{\mathcal{F}_A}(\mathcal{D}_S, \mathcal{D}_T) - \epsilon \leq \tau(A; \mathcal{D}_S, \mathcal{D}_T) \leq d_{\mathcal{F}_A}(\mathcal{D}_S, \mathcal{D}_T)$.*

**Interpretation:** In the above theorem, we show that given any $A, \mathcal{D}_S$, and the marginal $\mathcal{D}^{\mathcal{X}}$, there exists some conditional distributions of $y|x$ such that by composing it with the given $\mathcal{D}^{\mathcal{X}}$ we have a distribution $\mathcal{D}_T$ to make the bound in Theorem 3.1 $\epsilon$-tight. Note that $\epsilon$ is the difference between the source loss and the its infimum, i.e., with a good enough algorithm $A$, the $\epsilon$ could be arbitrarily small.

### 3.3 Generalization Upper Bound of the Relative Domain Transferability

Here we investigate the proposed theory on relative transferability with finite samples. For a distribution $\mathcal{D} \in \mathcal{P}_{\mathcal{X} \times \mathcal{Y}}$, we denote its empirical distribution with $n$ samples as $\widehat{\mathcal{D}}^n$. That being said,

$$\ell_{\widehat{\mathcal{D}}^n}(g \circ f) = \mathbb{E}_{(x,y) \sim \widehat{\mathcal{D}}^n}[\ell(g \circ f(x), y)] = \frac{1}{n} \sum_{i=1}^{n} \ell(g \circ f(x_i), y_i),$$

where $(x_i, y_i)$ are i.i.d. samples from $\mathcal{D}$. Therefore, given two distributions $\mathcal{D}_S, \mathcal{D}_T \in \mathcal{P}_{\mathcal{X} \times \mathcal{Y}}$, the empirical $(\mathcal{G}, \mathcal{F})$-pseudometric between them is $d_{\mathcal{G}, \mathcal{F}}(\widehat{\mathcal{D}}_S^n, \widehat{\mathcal{D}}_T^n)$.

Note that $d_{\mathcal{G}, \mathcal{F}}$ is not only a pseudometric of distributions, but also a complexity measure, and we will first connect it with the Rademacher complexity.

**Definition 3** (Empirical Rademacher Complexity (Bartlett & Mendelson, 2002; Koltchinskii, 2001))**.** *Denote the loss function class induced by $\mathcal{G}, \mathcal{F}$ as*

$$\mathcal{L}_{\mathcal{G}, \mathcal{F}} := \{h_{g,f} : \mathcal{X} \times \mathcal{Y} \to \mathbb{R}_+ \mid g \in \mathcal{G}, f \in \mathcal{F}\}, \quad where \ h_{g,f}(x, y) := \ell(g \circ f(x), y).$$

*Given an empirical distribution $\widehat{\mathcal{D}}^n$ (i.e., $n$ data samples), the Rademacher complexity of it is*

$$\operatorname{Rad}_{\widehat{\mathcal{D}}^n}(\mathcal{L}_{\mathcal{G}, \mathcal{F}}) := \frac{1}{n} \mathbb{E}_{\boldsymbol{\xi}} \left[ \sup_{h \in \mathcal{L}_{\mathcal{G}, \mathcal{F}}} \sum_{i=1}^n \xi_i h(x_i, y_i) \right],$$

*where $\boldsymbol{\xi} \in \mathbb{R}^n$ are Rademacher variables, i.e., each $\xi_i$ is i.i.d. uniformly distributed on $\{-1, 1\}$.*

We can see that if there is a $\mathcal{F}' \subseteq \mathcal{F}$, then $\operatorname{Rad}_{\widehat{\mathcal{D}}^n}(\mathcal{L}_{\mathcal{G}, \mathcal{F}'}) \leq \operatorname{Rad}_{\widehat{\mathcal{D}}^n}(\mathcal{L}_{\mathcal{G}, \mathcal{F}})$. With the above definitions, we have the following lemma connecting the $(\mathcal{G}, \mathcal{F})$-pseudometric to Rademacher complexity.

**Lemma 3.1.** *Assuming the individual loss function $\ell : \mathcal{Y} \times \mathcal{Y} \to [0, c]$, given any distribution $\mathcal{D} \in \mathcal{P}_{\mathcal{X} \times \mathcal{Y}}$ and $\forall \delta > 0$, with probability $\geq 1 - \delta$ we have*

$$d_{\mathcal{G}, \mathcal{F}}(\mathcal{D}, \widehat{\mathcal{D}}^n) \leq 2\operatorname{Rad}_{\widehat{\mathcal{D}}^n}(\mathcal{L}_{\mathcal{G}, \mathcal{F}}) + 3c\sqrt{\frac{\ln(4/\delta)}{2n}}.$$

Therefore, denoting again $d_{\mathcal{F}_A}$ as $d_{\mathcal{G}, \mathcal{F}_A}$, the empirical version of Theorem 3.1 is as follows.

**Theorem 3.3.** *Assuming the individual loss function $\ell : \mathcal{Y} \times \mathcal{Y} \to [0, c]$, given $\forall \mathcal{D}_S, \mathcal{D}_T \in \mathcal{P}_{\mathcal{X} \times \mathcal{Y}}$, for $\forall \delta > 0$ with probability $\geq 1 - \delta$ we have*

$$\tau(A; \widehat{\mathcal{D}}_S^n, \mathcal{D}_T) \leq d_{\mathcal{F}_A}(\widehat{\mathcal{D}}_S^n, \widehat{\mathcal{D}}_T^n) + 2\operatorname{Rad}_{\widehat{\mathcal{D}}_T^n}(\mathcal{L}_{\mathcal{G}, \mathcal{F}_A}) + 4\operatorname{Rad}_{\widehat{\mathcal{D}}_S^n}(\mathcal{L}_{\mathcal{G}, \mathcal{F}_A}) + 9c\sqrt{\frac{\ln(8/\delta)}{2n}}.$$

We can see that a smaller feature extractor function class $\mathcal{F}_A$ implies both a smaller $d_{\mathcal{F}_A}$ and the Rademacher complexity. Therefore, the monotone relation between the regularization strength and the upper bound on the relative domain transferability loss also holds for the empirical settings.

Other than direct regularization, empirically we find that the transferability is also related to the use of data augmentation. Can we explain such phenomena from the view of regularization again? We discuss this question next.

### 3.4 WHEN CAN DATA AUGMENTATION BE VIEWED AS REGULARIZATION?

In this subsection, we discuss the connections between data augmentation (DA) and regularization. We present the results and their interpretation in this subsection, while deferring the detailed discussion to the Section B in the appendix.

Empirical research has shown evidence of the regularization effect of DA (Hernández-García & König, 2018a;b). However, there is a lack of theoretical understanding on when can data augmentation be viewed as regularization in general. In an attempt to address this question, we consider a general DA setting of affine transformation (Perez & Wang, 2017) with parameters $(W_\star, b_\star)$ whose distribution represents specific DA.

**General Settings.** We consider the fine-tuning function $g : \mathbb{R}^d \to \mathbb{R}$ as a linear layer, which will be concatenated to the feature extractor $f : \mathbb{R}^m \to \mathbb{R}^d$. Given a model $g \circ f$, we use the squared loss $\ell(g \circ f(x), y) = (g \circ f(x) - y)^2$, and accordingly apply second-order Taylor expansion to the objective function to study the effect of data augmentation.

**DA categories.** We discuss two categories of DA, *feature-level DA* and *data-level DA*. Feature-level DA (Wong et al., 2016; DeVries & Taylor, 2017) requires the transformation to be performed in the learned feature space: given a data sample $x \in \mathbb{R}^m$ and a feature extractor $f$, the augmented feature is $W_\star f(x) + b_\star$ where $W_\star \in \mathbb{R}^{d \times d}, b_\star \in \mathbb{R}^d$ are sampled from a distribution. On the other hand,

data-level DA requires the transformation to be performed in the input space: given a data sample $x$, the augmented sample is $W_\star x + b_\star$ where $W_\star \in \mathbb{R}^{m \times m}, b_\star \in \mathbb{R}^m$ are sampled from a distribution.

**Intuition on sufficient conditions**. For either the feature-level or the data-level DA, the intuitions given by our analysis are similar. Our results (Theorem B.1&B.2) suggest that the following conditions indicate the regularization effects of a data augmentation: 1) $\mathbb{E}_{W_\star}[W_\star] = \mathbb{I}$; 2) $\mathbb{E}_{b_\star}[b_\star] = \vec{0}$; 3) $W_\star$ and $b_\star$ are independent, where $\mathbb{I}$ is the identity matrix and $\vec{0}$ is the zero vector; 4) $W_\star$ is not a constant if it is the feature-level DA; 5) DA is of a small magnitude if it is the data-level DA.

**Empirical verification.** Combining with Theorem 3.3, it suggests that DA satisfying the conditions above may improve the relative domain transferability. In fact, it matches the empirical observations in Section 4. Concretely, *1) Gaussian noise* *satisfies* the four conditions, and empirically the Gaussian noise improves domain transferability while robustness decreases a bit (Figure 5); *2) Rotation*, which rotates input image with a predefined fixed angle with predefined fixed probability, *violates* $\mathbb{E}_{W_\star}[W_\star] = \mathbb{I}$, and empirically the rotation barely affects domain transferability (Figure 14 in Appendix D.3); *3) Translation*, which moves the input image for a predefined distance along a pre-selected axis with fixed probability, *violates* $\mathbb{E}_{b_\star}[b_\star] = \vec{0}$, and empirically the translation distance barely co-relates to the domain transferability (Figure 14 in Appendix D.3).

## 4 EVALUATION

### 4.1 EXPERIMENTAL SETTING

**Source model training** We train our model on two source domains: CIFAR-10 and ImageNet. Unless specified, we will use the training setting as follows[1]. For CIFAR-10, we train the model with 200 epochs using the momentum SGD optimizer with momentum 0.9, weight decay 0.0005, an initial learning rate 0.1 which decays by a factor of 10 at the 100-th and 150-th epoch. For ImageNet, we train the model with 90 epochs using the momentum SGD optimizer with momentum 0.9, weight decay 0.0001, an initial learning rate 0.1 which decays by a factor of 10 at the 30-th and 60-th epoch. We use the standard cross entropy loss denote as $L_{CE}(h_s, x, y)$, where $h_s = g_s \circ f$ is the trained model and $x, y$ are the input and label respectively. To evaluate the robustness on the source domain, we follow the evaluation setting in Ilyas et al. (2019) and perform the PGD attack with 20 steps using $\epsilon = 0.25$. We also evaluate the robustness using AutoAttack in Appendix D.4. For both tasks we use ResNet-18 as the model structure. We provide results of other model structures in appendix D.2.

**Domain Transferability** We evaluate the transferability from CIFAR-10 to SVHN and from ImageNet to CIFAR-10. For the ImageNet transferability, we focus on CIFAR as the target domain, since it is the domain that is the most positively correlated with robustness as shown in Salman et al. (2020). We evaluate the fixed-feature transfer where only the last fully-connected layer is fine-tuned following our theoretical framework. We fine-tune the last layer with 40 epochs using momentum SGD optimizer with momentum 0.9, weight decay 0.0005, an initial learning rate 0.01 which decays by a factor of 10 at the 20-th and 30-th epoch. To mitigate the impact of benign accuracy, we evaluate the *relative domain transfer accuracy* (DT Acc) as follows. Let $acc_{src}$ and $acc_{tgt}$ be the accuracy of the a fine-tuned model on source and target domain, and $acc_{src}^v$ and $acc_{tgt}^v$ be the accuracy of vanilla model (*i.e.*, models trained with standard setting) on source and target domain, then the relative DT accuracy is defined as:

$$\text{DT Acc} = (acc_{tgt} - acc_{src}) - (acc_{tgt}^v - acc_{src}^v)$$

We also provide the results of absolute DT accuracy in Appendix D.1.

### 4.2 RELATIONSHIP BETWEEN ROBUSTNESS AND TRANSFERABILITY UNDER CONTROLLABLE CONDITIONS

We train the model under different controllable conditions to validate our analysis. In particular, we train the methods by controlling different regularization or data augmentations to evaluate the change of model robustness and transferability.

---

[1]These settings are inherited from the standard training algorithms for CIFAR-10 (`https://github.com/kuangliu/pytorch-cifar`) and ImageNet (`https://github.com/pytorch/examples/tree/master/imagenet`).

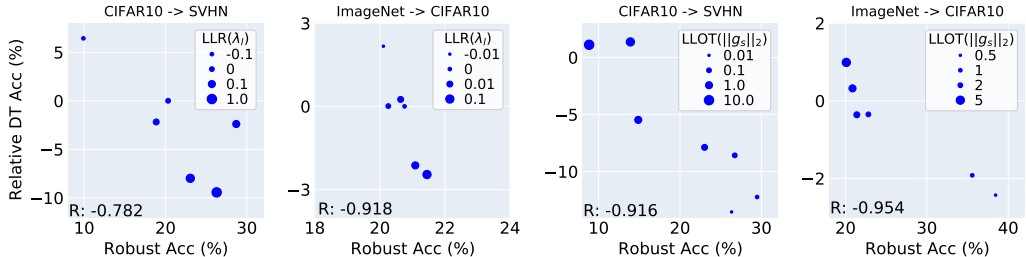

Figure 3: Relationship between robustness and transferability under different norms of last layer, via training with last-layer regularization (LLR) and last-layer orthogonalization (LLOT).

**Controlling the norm of last layer** As shown in our framework, domain transferability is related with the regularization on the feature extractor. Here we regularize the transferability by controlling the norm of last linear layer $g_s$. Intuitively, when we force the norm of $g_s$ to be large during training, the corresponding norm of $f$ will be regularized to be small. We use two approaches to control the last layer norm:

- Last-layer regularization (LLR): we impose a strong l2-regularizer with parameter $\lambda_l$ specifically on the weight of $g_s$ and therefore our training loss becomes: $L_{LLR}(h_s, x, y) = L_{CE}(h_s, x, y) + \lambda_l \cdot ||g_s||_F$, where $||g_s||_F$ is the frobenius norm of the weight matrix of $g_s$.
- Last-layer orthogonal training (LLOT): we directly control the l2-norm of $g_s$ with orthogonal training (Huang et al. (2020)). The orthogonal training will enforce the weight to become a 1-norm matrix and we multiply a constant to obtain the desired norm $||g_s||_2$.

The result of LLR and LLOT are shown in Figure 3. We observe that when we regularize the norm of last layer to be large (i.e. smaller $\lambda$ in LLR and larger $||g_s||_2$ in LLOT), the domain transferability will increase while the model robustness will decrease (their negative correlation is significant with Pearson's coefficient around $-0.9$). This is because larger last layer norm will produce a feature extractor $f$ with smaller norm, which, according to our analysis, leads to a better domain transferability. On the other hand, the model $g_s \circ f$ will have a larger norm and therefore becomes less robust under adversarial attacks.

**Controlling the norm of feature extractor** We directly regularize the feature extractor $f$ and check the impact on domain transferability. We implement two regularization as follows:

- Jacobian regularization (JR): we follow the approach in Hoffman et al. (2019) to apply JR on the feature extractor. Given model $h_s = g_s \circ f$, the training loss becomes: $L_{JR}(h_s, x, y) = L_{CE}(h_s, x, y) + \lambda_j \cdot ||J(f, x)||_F^2$, where $J(f, x)$ denotes the Jacobian matrix of $f$ on $x$ and $||\cdot||_F$ is the frobenius norm.
- Weight Decay (WD): we impose a strong weight decay with factor $\lambda_w$ on the feature extractor $f$ during training. This is equivalent to imposing a strong l2-regularizer with factor $\lambda_w$ on the feature extractor (excluding the last layer).

The results under JR and WD are shown in Figure 4. We observe that with larger regularization on the feature extractor, the model shows higher domain transferability, which matches our analysis. Meanwhile, the robustness decreases significantly with large regularizer. This is because a large regularization will harm the model performance on source domain and lead to low model robustness.

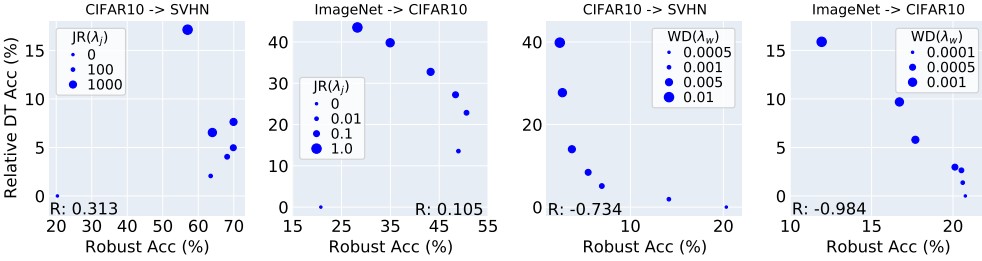

Figure 4: Relationship between robustness and transferability when we regularize the feature extractor with Jacobian Regularization (JR) and weight decay (WD).

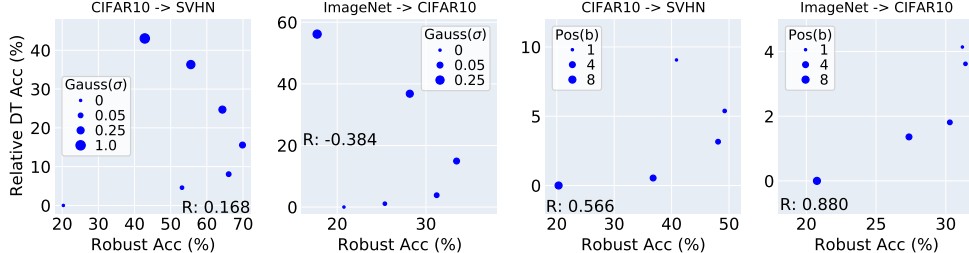

Figure 5: Relationship between robustness and transferability when we use Gaussian noise (Gauss) and posterize (Pos) as data augmentations.

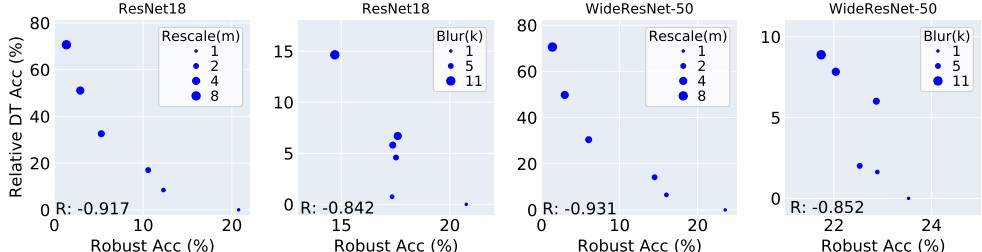

Figure 6: Relationship between robustness and transferability on ImageNet when we use rescale and blur as data augmentations.

**Data augmentation** As shown in Section 3.4, data augmentation can be viewed as a type of regularization during training and thus affects the domain transferability. Here we consider both noise dependent and independent data augmentations.

**Noise-dependent data augmentation** We include two noise-dependent augmentations:

- Gaussian Noise data augmentation (Gauss): we add zero-mean Gaussian noise with variance $\sigma^2$ to the input image.

- Posterize (Pos): we truncate each channel of one pixel value into $b$ bits (originally they are 8 bits).

The results of Gauss and Pos are shown in Figure 5. We observe that the domain transferability of trained model keeps improving with larger data augmentation, which matches our theory. The robustness also benefits from a small data augmentation, but decreases when it becomes large.

**Resolution-related (noise-independent) data augmentation.** Specifically, for ImageNet to CIFAR-10 transferability, we consider two resolution-related data augmentations. The intuition is that when the target domain has a lower resolution than the source domain (ImageNet is $224 \times 224$ while CIFAR-10 is $32 \times 32$), the data augmentations that down-sample the inputs during the training on source domain will help transferability. We consider the below resolution-related augmentations:

- Rescale: we rescale the input to be $m$ times smaller (*i.e.*, shape ImageNet as $(224/m) \times (224/m)$) and then rescale them back to the original size.

- Blur: we apply Gaussian blurring with kernel size $k$ on the input. The Gaussian kernel is created with a standard deviation randomly sampled from $[0.1, 2.0]$.

We show the results of rescaling and blurring in Figure 6. The experiments are evaluated only for ImageNet to CIFAR-10, and we include the results of both ResNet18 (the default model) and WideResNet50. We can see that these data augmentations help with transferability to target domain, although the robustness on the source domain decreases since these augmentations do not include any robustness-related operations.

## 5 CONCLUSIONS

In this work, we theoretically analyze the sufficient conditions for domain transferability based on the view of function class regularization. We also conduct experiments to verify our claims and observe some counterexamples that shows a negative correlation between robustness and domain transferability. These results are helpful in the domain generalization of machine learning models.

ETHICS STATEMENT

Our work focuses on theoretically and empirically studying the domain transferability of a machine learning model. All the datasets and packages we use are open-sourced. We do not have ethical concerns in our paper.

REPRODUCIBILITY STATEMENT

We have tried our best to provide training details to facilitate reproducing our results. In Section 4.1 we provide detailed results on how to train the model and how to transfer the trained model to target domain, as well as how we evaluate our model. We also upload the zip file of our code with the submission. We will open-source our code once accepted.

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

# Appendix

## A  PROOFS

**Proposition A.1** (Proposition 3.1 Restated). *Given the problem defined in subsection 3.1, $f_c^{\mathcal{D}_S}$ is a minimizer of equation 1. Moreover, if $c \geq c' \geq 0$, then the relative domain transfer loss $\ell_{\mathcal{D}_T}(f_c^{\mathcal{D}_S}) - \ell_{\mathcal{D}_S}(f_c^{\mathcal{D}_S}) \geq \ell_{\mathcal{D}_T}(f_{c'}^{\mathcal{D}_S}) - \ell_{\mathcal{D}_S}(f_{c'}^{\mathcal{D}_S})$.*

*Proof.* Recall that $\ell_{\mathcal{D}_S}(f) = \|f - y_S\|_{\mathcal{D}}$, $\ell_{\mathcal{D}_T}(f) = \|f - y_T\|_{\mathcal{D}}$ and $f_c^{\mathcal{D}_S} := y_S \min\{1, \frac{c}{\|y_S\|_{\mathcal{D}}}\}$. First, let's verify that

$$f_c^{\mathcal{D}_S} \in \arg\min_{f \in \mathcal{F}} \ell_{\mathcal{D}_S}(f), \quad \text{s.t. } \|f\|_{\mathcal{D}} \leq c.$$

If $c \geq \|y_S\|_{\mathcal{D}}$, then $f_c^{\mathcal{D}_S} := y_S$ achieves the minimum. If $c < \|y_S\|_{\mathcal{D}}$, then for any $f \in \mathcal{F} : \|f\|_{\mathcal{D}} \leq c$, we have

$$\ell_{\mathcal{D}_S}(f) = \|f - y_S\|_{\mathcal{D}} \geq \|y_S\|_{\mathcal{D}} - \|f\|_{\mathcal{D}} \geq \|y_S\|_{\mathcal{D}} - c$$
$$= \|(1 - \tfrac{c}{\|y_S\|_{\mathcal{D}}})y_S\|_{\mathcal{D}} = \|y_S - \tfrac{c}{\|y_S\|_{\mathcal{D}}}y_S\|_{\mathcal{D}} = \ell_{\mathcal{D}_S}(f_c^{\mathcal{D}_S}).$$

Therefore, $f_c^{\mathcal{D}_S}$ indeed achieves the minimum.

Now, let's prove the proposition. For any $c \geq \|y_S\|_{\mathcal{D}}$, we have $\ell_{\mathcal{D}_S}(f_c^{\mathcal{D}_S}) = 0$ and $\ell_{\mathcal{D}_T}(f_c^{\mathcal{D}_S})$ is a constant. Therefore, there is no difference for all $c \geq \|y_S\|_{\mathcal{D}}$, and the proposition holds for $c \geq c' \geq \|y_S\|_{\mathcal{D}}$. Then, We only need to verify the case for $\|y_S\| \geq c \geq c'$:

$$\ell_{\mathcal{D}_S}(f_{c'}^{\mathcal{D}_S}) - \ell_{\mathcal{D}_S}(f_c^{\mathcal{D}_S}) = c - c' = \|\tfrac{c}{\|y_S\|_{\mathcal{D}}}y_S - \tfrac{c'}{\|y_S\|_{\mathcal{D}}}y_S\|_{\mathcal{D}}$$
$$= \|f_{c'}^{\mathcal{D}_S} - f_c^{\mathcal{D}_S}\|_{\mathcal{D}} = \|f_{c'}^{\mathcal{D}_S} - y_T + y_T - f_c^{\mathcal{D}_S}\|_{\mathcal{D}}$$
$$\geq |\|f_{c'}^{\mathcal{D}_S} - y_T\|_{\mathcal{D}} - \|y_T - f_c^{\mathcal{D}_S}\|_{\mathcal{D}}|$$
$$\geq \|f_{c'}^{\mathcal{D}_S} - y_T\|_{\mathcal{D}} - \|y_T - f_c^{\mathcal{D}_S}\|_{\mathcal{D}}$$
$$= \ell_{\mathcal{D}_T}(f_{c'}^{\mathcal{D}_S}) - \ell_{\mathcal{D}_T}(f_c^{\mathcal{D}_S}).$$

Rearranging the above inequality gives the proposition. $\square$

**Proposition A.2** (Proposition 3.2 Restated). *$d_{\mathcal{G},\mathcal{F}}(\cdot, \cdot) : \mathcal{P}_{\mathcal{X} \times \mathcal{Y}} \times \mathcal{P}_{\mathcal{X} \times \mathcal{Y}} \to \mathbb{R}_+$ satisfies the following three properties.*

1. *(Symmetry) $d_{\mathcal{G},\mathcal{F}}(\mathcal{D}_S, \mathcal{D}_T) = d_{\mathcal{G},\mathcal{F}}(\mathcal{D}_T, \mathcal{D}_S)$.*

2. *(Triangle Inequality) For $\forall \mathcal{D}' \in \mathcal{P}_{\mathcal{X} \times \mathcal{Y}}$: $d_{\mathcal{G},\mathcal{F}}(\mathcal{D}_S, \mathcal{D}_T) \leq d_{\mathcal{G},\mathcal{F}}(\mathcal{D}_S, \mathcal{D}') + d_{\mathcal{G},\mathcal{F}}(\mathcal{D}', \mathcal{D}_T)$.*

3. *(Weak Zero Property) For $\forall \mathcal{D} \in \mathcal{P}_{\mathcal{X} \times \mathcal{Y}}$: $d_{\mathcal{G},\mathcal{F}}(\mathcal{D}, \mathcal{D}) = 0$.*

*Proof.* Recall that

$$d_{\mathcal{G},\mathcal{F}}(\mathcal{D}_S, \mathcal{D}_T) := \sup_{f \in \mathcal{F}} |\inf_{g \in \mathcal{G}} \ell_{\mathcal{D}_S}(g \circ f) - \inf_{g \in \mathcal{G}} \ell_{\mathcal{D}_T}(g \circ f)|.$$

We can see that the symmetry and weak zero property are obvious. For triangle inequality, given $\forall \mathcal{D}' \in \mathcal{P}_{\mathcal{X} \times \mathcal{Y}}$:

$$d_{\mathcal{G},\mathcal{F}}(\mathcal{D}_S, \mathcal{D}_T) = \sup_{f \in \mathcal{F}} |\inf_{g \in \mathcal{G}} \ell_{\mathcal{D}_S}(g \circ f) - \inf_{g \in \mathcal{G}} \ell_{\mathcal{D}_T}(g \circ f)|$$
$$= \sup_{f \in \mathcal{F}} |\inf_{g \in \mathcal{G}} \ell_{\mathcal{D}_S}(g \circ f) - \inf_{g \in \mathcal{G}} \ell_{\mathcal{D}'}(g \circ f) + \inf_{g \in \mathcal{G}} \ell_{\mathcal{D}'}(g \circ f) - \inf_{g \in \mathcal{G}} \ell_{\mathcal{D}_T}(g \circ f)|$$
$$\leq \sup_{f \in \mathcal{F}} (|\inf_{g \in \mathcal{G}} \ell_{\mathcal{D}_S}(g \circ f) - \inf_{g \in \mathcal{G}} \ell_{\mathcal{D}'}(g \circ f)| + |\inf_{g \in \mathcal{G}} \ell_{\mathcal{D}'}(g \circ f) - \inf_{g \in \mathcal{G}} \ell_{\mathcal{D}_T}(g \circ f)|)$$
$$\leq \sup_{f \in \mathcal{F}} |\inf_{g \in \mathcal{G}} \ell_{\mathcal{D}_S}(g \circ f) - \inf_{g \in \mathcal{G}} \ell_{\mathcal{D}'}(g \circ f)| + \sup_{f \in \mathcal{F}} |\inf_{g \in \mathcal{G}} \ell_{\mathcal{D}'}(g \circ f) - \inf_{g \in \mathcal{G}} \ell_{\mathcal{D}_T}(g \circ f)|$$
$$= d_{\mathcal{G},\mathcal{F}}(\mathcal{D}_S, \mathcal{D}') + d_{\mathcal{G},\mathcal{F}}(\mathcal{D}', \mathcal{D}_T).$$

$\square$

**Proposition A.3.** *Denote the function class*

$$\mathcal{L}_{\mathcal{G},\mathcal{F}} := \{h_{g,f} : \mathcal{X} \times \mathcal{Y} \to \mathbb{R}_+ \mid g \in \mathcal{G}, f \in \mathcal{F}\}, \quad \text{where } h_{g,f}(x,y) := \ell(g \circ f(x), y).$$

*Let $d : \mathcal{X} \times \mathcal{Y} \to \mathbb{R}_+$ be a metric on $\mathcal{X} \times \mathcal{Y}$, and assume $\forall h \in \mathcal{L}_{\mathcal{G},\mathcal{F}}$ is L-Lipschitz continuous with respect to the metric d. Then, we have*

$$d_{\mathcal{F}_A}(\mathcal{D}_S, \mathcal{D}_T) \le L \cdot W(\mathcal{D}_S, \mathcal{D}_T),$$

*where $W(\mathcal{D}_S, \mathcal{D}_T)$ is the Wasserstein distance:*

$$W(\mathcal{D}_S, \mathcal{D}_T) = \sup_{\phi:\mathcal{X}\times\mathcal{Y}\to\mathbb{R}} \mathbb{E}_{(x,y)\sim\mathcal{D}_S}[\phi(x,y)] - \mathbb{E}_{(x,y)\sim\mathcal{D}_T}[\phi(x,y)] \quad \text{s.t. } \phi \text{ is 1-Lipschitz.}$$

*Proof.* Recall that

$$d_{\mathcal{G},\mathcal{F}}(\mathcal{D}_S, \mathcal{D}_T) := \sup_{f\in\mathcal{F}} |\inf_{g\in\mathcal{G}} \ell_{\mathcal{D}_S}(g \circ f) - \inf_{g\in\mathcal{G}} \ell_{\mathcal{D}_T}(g \circ f)|.$$

By the definition of inf, for $\forall \epsilon > 0$ there exist $g_{S,\epsilon}, g_{T,\epsilon} \in \mathcal{G}$ such that

$$\inf_{g\in\mathcal{G}} \ell_{\mathcal{D}_S}(g \circ f_\epsilon) \ge \ell_{\mathcal{D}_S}(g_{S,\epsilon} \circ f_\epsilon) - \epsilon$$
$$\inf_{g\in\mathcal{G}} \ell_{\mathcal{D}_T}(g \circ f_\epsilon) \ge \ell_{\mathcal{D}_T}(g_{T,\epsilon} \circ f_\epsilon) - \epsilon.$$

By the definition of sup, there exists $f_\epsilon \in \mathcal{F}$ such that

$$d_{\mathcal{G},\mathcal{F}}(\mathcal{D}_S, \mathcal{D}_T) \le |\inf_{g\in\mathcal{G}} \ell_{\mathcal{D}_S}(g \circ f_\epsilon) - \inf_{g\in\mathcal{G}} \ell_{\mathcal{D}_T}(g \circ f_\epsilon)| + \epsilon$$
$$= \max\{\inf_{g\in\mathcal{G}} \ell_{\mathcal{D}_S}(g \circ f_\epsilon) - \inf_{g\in\mathcal{G}} \ell_{\mathcal{D}_T}(g \circ f_\epsilon), \inf_{g\in\mathcal{G}} \ell_{\mathcal{D}_T}(g \circ f_\epsilon) - \inf_{g\in\mathcal{G}} \ell_{\mathcal{D}_S}(g \circ f_\epsilon)\} + \epsilon$$
$$\le \max\{\ell_{\mathcal{D}_S}(g_{T,\epsilon} \circ f_\epsilon) - \inf_{g\in\mathcal{G}} \ell_{\mathcal{D}_T}(g \circ f_\epsilon), \ell_{\mathcal{D}_T}(g_{S,\epsilon} \circ f_\epsilon) - \inf_{g\in\mathcal{G}} \ell_{\mathcal{D}_S}(g \circ f_\epsilon)\} + \epsilon$$
$$\le \max\{\ell_{\mathcal{D}_S}(g_{T,\epsilon} \circ f_\epsilon) - \ell_{\mathcal{D}_T}(g_{T,\epsilon} \circ f_\epsilon), \ell_{\mathcal{D}_T}(g_{S,\epsilon} \circ f_\epsilon) - \ell_{\mathcal{D}_S}(g_{S,\epsilon} \circ f_\epsilon)\} + 2\epsilon.$$
$$(2)$$

Let's first consider the first term in the $\max\{\cdot, \cdot\}$ above.

$$\ell_{\mathcal{D}_S}(g_{T,\epsilon} \circ f_\epsilon) - \ell_{\mathcal{D}_T}(g_{T,\epsilon} \circ f_\epsilon)$$
$$= L \cdot (\tfrac{1}{L}\ell_{\mathcal{D}_S}(g_{T,\epsilon} \circ f_\epsilon) - \tfrac{1}{L}\ell_{\mathcal{D}_T}(g_{T,\epsilon} \circ f_\epsilon))$$
$$= L \cdot (\mathbb{E}_{(x,y)\sim\mathcal{D}_S}[\tfrac{1}{L}\ell(g_{T,\epsilon} \circ f_\epsilon(x), y)] - \mathbb{E}_{(x,y)\sim\mathcal{D}_T}[\tfrac{1}{L}\ell(g_{T,\epsilon} \circ f_\epsilon(x), y)])$$
$$\le L \cdot W(\mathcal{D}_S, \mathcal{D}_T),$$

where the inequality is due to that both $\frac{1}{L}\ell(g_{S,\epsilon} \circ f_\epsilon(x), y)$ and $\frac{1}{L}\ell(g_{T,\epsilon} \circ f_\epsilon(x), y)$ are 1-Lipschitz w.r.t. $(x, y)$ and the metric $d$.

Similarly, we also have

$$\ell_{\mathcal{D}_T}(g_{S,\epsilon} \circ f_\epsilon) - \ell_{\mathcal{D}_S}(g_{S,\epsilon} \circ f_\epsilon) \le L \cdot W(\mathcal{D}_S, \mathcal{D}_T).$$

Therefore, equation 2 implies

$$d_{\mathcal{G},\mathcal{F}}(\mathcal{D}_S, \mathcal{D}_T) \le L \cdot W(\mathcal{D}_S, \mathcal{D}_T) + 2\epsilon.$$

Letting $\epsilon \to 0$ completes the proof.

$$\square$$

**Proposition A.4.** *Consider multi-class classification where $\mathcal{Y} = [k]$ for some $k \ge 2$. Define the loss function $\ell$ as*

$$\ell(g \circ f(x), y) = \mathbb{1}\{\arg\max_{j\in[k]}(g \circ f(x))_j \ne y\}$$

*Let $\delta_{TV}(D_S, D_T)$ denote the total variation distance. Then we have*

$$d_{\mathcal{F}_A}(\mathcal{D}_S, \mathcal{D}_T) \le \delta_{TV}(\mathcal{D}_S, \mathcal{D}_T)$$

*Proof.* Fix $f \in \mathcal{F}$. By the definition of inf, there exists $g_{T,\epsilon}$ such that

$$
\begin{aligned}
&\left| \inf_{g \in \mathcal{G}} \ell_{\mathcal{D}_S}(g \circ f) - \inf_{g \in \mathcal{G}} \ell_{\mathcal{D}_T}(g \circ f) \right| \\
&\leq \left| \inf_{g \in \mathcal{G}} \ell_{\mathcal{D}_S}(g \circ f) - \ell_{\mathcal{D}_T}(g_{T,\epsilon} \circ f) \right| + \epsilon \\
&\leq |\ell_{\mathcal{D}_S}(g_{T,\epsilon} \circ f) - \ell_{\mathcal{D}_T}(g_{T,\epsilon} \circ f)| + \epsilon \\
&= |\mathbb{E}_{(x,y) \sim \mathcal{D}_S}[\ell(g_{T,\epsilon} \circ f(x), y)] - \mathbb{E}_{(x,y) \sim \mathcal{D}_T}[\ell(g_{T,\epsilon} \circ f(x), y)]| + \epsilon \\
&= |\mathbb{P}_{(x,y) \sim \mathcal{D}_S}[\mathbb{1}\{\arg\max_{j \in [k]}(g_{T,\epsilon} \circ f(x))_j \neq y\}] - \mathbb{P}_{(x,y) \sim \mathcal{D}_T}[\mathbb{1}\{\arg\max_{j \in [k]}(g_{T,\epsilon} \circ f(x))_j \neq y\}]| + \epsilon
\end{aligned}
\tag{3}
$$

Let $A$ be the event such that $A = \{(x,y) : \arg\max_{j \in [k]}(g_{T,\epsilon} \circ f(x))_j \neq y\}$. Then we can write equation 3 as

$$
\begin{aligned}
(3) &= |\mathbb{P}_{(x,y) \sim \mathcal{D}_S}[A] - \mathbb{P}_{(x,y) \sim \mathcal{D}_T}[A]| + \epsilon \\
&\leq \sup_{B} |\mathbb{P}_{(x,y) \sim \mathcal{D}_S}[B] - \mathbb{P}_{(x,y) \sim \mathcal{D}_T}[B]| + \epsilon \\
&= \delta_{TV}(\mathcal{D}_S, \mathcal{D}_T) + \epsilon
\end{aligned}
$$

Send $\epsilon \to 0$. Noting that $f \in \mathcal{F}$ was arbitrary, apply sup to both sides gives us the desired inequality. □

Theorem 3.1 can be proved easily by definition.

**Theorem A.1** (Theorem 3.1 Restated). *Given a training algorithm A, for $\forall \mathcal{D}_S, \mathcal{D}_T \in \mathcal{P}_{\mathcal{X} \times \mathcal{Y}}$ we have*

$$
\tau(A; \mathcal{D}_S, \mathcal{D}_T) \leq d_{\mathcal{F}_A}(\mathcal{D}_S, \mathcal{D}_T),
$$

*or equivalently,* $\quad \inf_{g \in \mathcal{G}} \ell_{\mathcal{D}_T}(g \circ f_A^{\mathcal{D}_S}) \leq \ell_{\mathcal{D}_S}(g_A^{\mathcal{D}_S} \circ f_A^{\mathcal{D}_S}) + d_{\mathcal{F}_A}(\mathcal{D}_S, \mathcal{D}_T).$

*Proof.* By definition,

$$
\begin{aligned}
\tau(A; \mathcal{D}_S, \mathcal{D}_T) &= \inf_{g \in \mathcal{G}} \ell_{\mathcal{D}_T}(g \circ f_A^{\mathcal{D}_S}) - \ell_{\mathcal{D}_S}(g_A^{\mathcal{D}_S} \circ f_A^{\mathcal{D}_S}) \\
&\leq \inf_{g \in \mathcal{G}} \ell_{\mathcal{D}_T}(g \circ f_A^{\mathcal{D}_S}) - \inf_{g \in \mathcal{G}} \ell_{\mathcal{D}_S}(g \circ f_A^{\mathcal{D}_S}) \\
&\leq |\inf_{g \in \mathcal{G}} \ell_{\mathcal{D}_T}(g \circ f_A^{\mathcal{D}_S}) - \inf_{g \in \mathcal{G}} \ell_{\mathcal{D}_S}(g \circ f_A^{\mathcal{D}_S})| \\
&\leq \sup_{f \in \mathcal{F}_A} |\inf_{g \in \mathcal{G}} \ell_{\mathcal{D}_T}(g \circ f) - \inf_{g \in \mathcal{G}} \ell_{\mathcal{D}_S}(g \circ f)| \\
&= d_{\mathcal{F}_A}(\mathcal{D}_S, \mathcal{D}_T).
\end{aligned}
$$

□

To prove Theorem 3.2, we first prove the following interesting lemma.

**Lemma A.1.** *Let $\mathcal{S}_r^{d-1} := \{y \in \mathbb{R}^d \mid \|y\|_2 = r\}$ denotes the $(d-1)$-dimensional sphere in $\mathbb{R}^d$ with radius $r > 0$. If a function $h : \mathcal{S}_r^{d-1} \to \mathbb{R}^d$ satisfies*

$$
\forall y \in \mathcal{S}_r^{d-1} : \quad \langle h(y), y \rangle < 0,
\tag{4}
$$

*then we have*

$$
\vec{0} \in \text{conv}(h(\mathcal{S}_r^{d-1})),
$$

*i.e., $\vec{0}$ is in the convex hull of $\{h(y) \mid y \in \mathcal{S}_r^{d-1}\}$.*

*Proof.* We assume that $\vec{0} \notin \text{conv}(h(\mathcal{S}_r^{d-1}))$ and prove by contradiction. Since $\vec{0} \notin \text{conv}(h(\mathcal{S}_r^{d-1}))$, we can find a hyperplane that separates $\vec{0}$ and the convex set $\text{conv}(h(\mathcal{S}_r^{d-1}))$. By the separating hyperplane theorem there exists a nonzero vector $v \in \mathbb{R}^d$ and $c \geq 0$ such that

$$
\forall z \in \text{conv}(h(\mathcal{S}_r^{d-1})) : \quad \langle z, v \rangle \geq c \geq 0.
\tag{5}
$$

We choose $y = r\boldsymbol{v}/\|\boldsymbol{v}\|_2$ and observe that $h(y) \in \text{conv}(h(\mathcal{S}_r^{d-1}))$. Hence, by equation 5 we have

$$\langle h(y), y \rangle \geq 0,$$

which contradicts to the condition of equation 4. Therefore, it must be that $\vec{0} \in \text{conv}(h(\mathcal{S}_r^{d-1}))$.

$\square$

**Theorem A.2** (Theorem 3.2 Restated). *Given any source distribution $\mathcal{D}_S \in \mathcal{P}_{\mathcal{X} \times \mathbb{R}^d}$, any fine-tuning function class $\mathcal{G}$ where $\mathcal{G}$ includes the zero function, and any training algorithm $A$, denote*

$$\epsilon := \ell_{\mathcal{D}_S}(g_A^{\mathcal{D}_S} \circ f_A^{\mathcal{D}_S}) - \inf_{g \in \mathcal{G}, f \in \mathcal{F}_A} \ell_{\mathcal{D}_S}(g \circ f).$$

*We assume some properties of the sample individual loss function $\ell : \mathbb{R}^d \times \mathbb{R}^d \to \mathbb{R}_+$: it is differentiable and strictly convex w.r.t. its first argument; $\ell(y, y) = 0$ for any $y \in \mathbb{R}^d$; and $\lim_{r \to \infty} \inf_{y : \|y\|_2 = r} \ell(\vec{0}, y) = \infty$. Then, for any distribution $\mathcal{D}^{\mathcal{X}}$ on $\mathcal{X}$, there exist some distributions $\mathcal{D}_T \in \mathcal{P}_{\mathcal{X} \times \mathcal{Y}}$ with its marginal on $\mathcal{X}$ being $\mathcal{D}^{\mathcal{X}}$ such that*

$$\tau(A; \mathcal{D}_S, \mathcal{D}_T) \leq d_{\mathcal{F}_A}(\mathcal{D}_S, \mathcal{D}_T) \leq \tau(A; \mathcal{D}_S, \mathcal{D}_T) + \epsilon.$$

*Proof.* The $\tau(A; \mathcal{D}_S, \mathcal{D}_T) \leq d_{\mathcal{F}_A}(\mathcal{D}_S, \mathcal{D}_T)$ is proved by Theorem 3.1, we only need to prove that there exists some $\mathcal{D}_T \in \mathcal{P}_{\mathcal{X} \times \mathcal{Y}}$ with its marginal on $\mathcal{X}$ being $\mathcal{D}^{\mathcal{X}}$ such that

$$d_{\mathcal{F}_A}(\mathcal{D}_S, \mathcal{D}_T) \leq \tau(A; \mathcal{D}_S, \mathcal{D}_T) + \epsilon = \inf_{g \in \mathcal{G}} \ell_{\mathcal{D}_T}(g \circ f_A^{\mathcal{D}_S}) - \inf_{g \in \mathcal{G}, f \in \mathcal{F}_A} \ell_{\mathcal{D}_S}(g \circ f).$$

We begin by observing that $\lim_{r \to \infty} \inf_{y : \|y\|_2 = r} \ell(\vec{0}, y) = \infty$, and thus there exists $r > 0$ such that

$$\forall y \in \mathcal{S}_r^{d-1} : \quad \ell(\vec{0}, y) \geq \ell_{\mathcal{D}_S}(\vec{0}) = \mathbb{E}_{(\boldsymbol{x}, y) \sim \mathcal{D}_S}[\ell(\vec{0}, y)], \tag{6}$$

where $\mathcal{S}_r^{d-1} := \{y \in \mathbb{R}^d \mid \|y\|_2 = r\}$ denotes the $(d-1)$-dimensional sphere with radius $r$. Note the we abuse the notion a bit to let $\vec{0}$ also denotes the zero function (i.e., maps all input to zero). Now, let us define at the following set

$$\mathcal{V} := \{\nabla_1 \ell(\vec{0}, y) \mid y \in \mathcal{S}_r^{d-1}\},$$

where $\nabla_1$ is taking the gradient w.r.t. the first argument of $\ell(\cdot, \cdot)$. By the strict convexity of $\ell(\cdot, y)$, we have

$$\ell(y, y) - \ell(\vec{0}, y) > \langle \nabla_1 \ell(\vec{0}, y), y \rangle.$$

Noting that $\ell(y, y) = 0$ is the unique minimum of $\ell(\cdot, y)$, we have $\ell(\vec{0}, y) > 0$. Accordingly,

$$\forall y \in \mathcal{S}_r^{d-1} : \quad 0 > -\ell(\vec{0}, y) > \langle \nabla_1 \ell(\vec{0}, y), y \rangle.$$

Having the above property, and noting that $\nabla_1 \ell(\vec{0}, \cdot) : \mathcal{S}_r^{d-1} \to \mathbb{R}^d$, we can invoke Lemma A.1 to see that

$$\vec{0} \in \text{conv}(\mathcal{V}).$$

Therefore, there exists $n$ points $\{y_i\}_{i=1}^n \subset \mathcal{S}_r^{d-1}$ such that

$$\vec{0} = \sum_{i=1}^n c_i \nabla_1 \ell(\vec{0}, y_i), \tag{7}$$

where $c_i > 0$ and $\sum_{i=1}^n c_i = 1$.

Therefore, we can define the target distribution $\mathcal{D}_T$ as the following. Given any $\boldsymbol{x} \sim \mathcal{D}^{\mathcal{X}}$, the distribution of $y$ conditioned on $\boldsymbol{x}$ is: $y = y_i$ with probability $c_i$. Now we verify the distribution $\mathcal{D}_T$ indeed makes the bound $\epsilon$-tight. Denote a strictly convex function $h : \mathbb{R}^d \to \mathbb{R}_+$ as the following

$$h(\cdot) := \sum_{i=1}^n c_i \ell(\cdot, y_i).$$

Since $h$ is strictly convex and $\nabla h(\vec{0}) = \vec{0}$ (equation 7), we can see that $h(\vec{0})$ achieves the unique global minimum of $h$ on $\mathbb{R}^d$.

Therefore, given the $\mathcal{D}_T$, for any $\forall f \in \mathcal{F}_A$ we have

$$
\begin{aligned}
\inf_{g \in \mathcal{G}} \ell_{\mathcal{D}_T}(g \circ f) &= \inf_{g \in \mathcal{G}} \mathbb{E}_{(\boldsymbol{x}, y) \sim \mathcal{D}_T}[\ell(g \circ f(\boldsymbol{x}), y)] \\
&= \inf_{g \in \mathcal{G}} \mathbb{E}_{\boldsymbol{x} \sim \mathcal{D}^{\mathcal{X}}} \left[ \sum_{i=1}^{n} c_i \ell(g \circ f(\boldsymbol{x}), y_i) \right] \\
&= \inf_{g \in \mathcal{G}} \mathbb{E}_{\boldsymbol{x} \sim \mathcal{D}^{\mathcal{X}}} [h(g \circ f(\boldsymbol{x}))] \\
&= h(\vec{0}) \qquad\qquad (\mathcal{G} \text{ contains the zero function}) \\
&= \sum_{i=1}^{n} c_i \ell(\vec{0}, y_i).
\end{aligned} \tag{8}
$$

Recall that $d_{\mathcal{F}_A}(\mathcal{D}_S, \mathcal{D}_T) = \sup_{f \in \mathcal{F}_A} |\inf_{g \in \mathcal{G}} \ell_{\mathcal{D}_T}(g \circ f) - \inf_{g \in \mathcal{G}} \ell_{\mathcal{D}_S}(g \circ f)|$, we can see that

$$
d_{\mathcal{F}_A}(\mathcal{D}_S, \mathcal{D}_T) = \sup_{f \in \mathcal{F}_A} | \sum_{i=1}^{n} c_i \ell(\vec{0}, y_i) - \inf_{g \in \mathcal{G}} \ell_{\mathcal{D}_S}(g \circ f)| \tag{9}
$$

By equation 6, for $\forall f \in \mathcal{F}_A$, we have

$$
\sum_{i=1}^{n} c_i \ell(\vec{0}, y_i) \geq \ell_{\mathcal{D}_S}(\vec{0}) = \ell_{\mathcal{D}_S}(\vec{0} \circ f) \geq \inf_{g \in \mathcal{G}} \ell_{\mathcal{D}_S}(g \circ f).
$$

Hence, we can continue as

$$
\begin{aligned}
(9) &= \sup_{f \in \mathcal{F}_A} \left( \sum_{i=1}^{n} c_i \ell(\vec{0}, y_i) - \inf_{g \in \mathcal{G}} \ell_{\mathcal{D}_S}(g \circ f) \right) = \sum_{i=1}^{n} c_i \ell(\vec{0}, y_i) - \inf_{g \in \mathcal{G}, f \in \mathcal{F}_A} \ell_{\mathcal{D}_S}(g \circ f) \\
&= \inf_{g \in \mathcal{G}} \ell_{\mathcal{D}_T}(g \circ f_A^{\mathcal{D}_S}) - \inf_{g \in \mathcal{G}, f \in \mathcal{F}_A} \ell_{\mathcal{D}_S}(g \circ f) \qquad\qquad \text{(by equation 8)} \\
&= \inf_{g \in \mathcal{G}} \ell_{\mathcal{D}_T}(g \circ f_A^{\mathcal{D}_S}) - \ell_{\mathcal{D}_S}(g_A^{\mathcal{D}_S} \circ f_A^{\mathcal{D}_S}) + \ell_{\mathcal{D}_S}(g_A^{\mathcal{D}_S} \circ f_A^{\mathcal{D}_S}) - \inf_{g \in \mathcal{G}, f \in \mathcal{F}_A} \ell_{\mathcal{D}_S}(g \circ f) \\
&= \tau(A; \mathcal{D}_S, \mathcal{D}_T) + \epsilon.
\end{aligned}
$$

Therefore, it holds that $d_{\mathcal{F}_A}(\mathcal{D}_S, \mathcal{D}_T) \leq \tau(A; \mathcal{D}_S, \mathcal{D}_T) + \epsilon$, and thus the theorem.

$\square$

**Lemma A.2** (Lemma 3.1 Restated). *Assuming the individual loss function $\ell : \mathcal{Y} \times \mathcal{Y} \to [0, c]$, given any distribution $\mathcal{D} \in \mathcal{P}_{\mathcal{X} \times \mathcal{Y}}$ and $\forall \delta > 0$, with probability $\geq 1 - \delta$ we have*

$$
d_{\mathcal{G}, \mathcal{F}}(\mathcal{D}, \widehat{\mathcal{D}}^n) \leq 2\text{Rad}_{\widehat{\mathcal{D}}^n}(\mathcal{L}_{\mathcal{G}, \mathcal{F}}) + 3c\sqrt{\frac{\ln(4/\delta)}{2n}}.
$$

*Proof.* Given any $\delta > 0$, $f \in \mathcal{F}$, $g \in \mathcal{G}$, $\mathcal{D} \in \mathcal{P}_{\mathcal{X} \times \mathcal{Y}}$, and taking any $h_{g,f} \in \mathcal{L}_{\mathcal{G}, \mathcal{F}}$ (Definition 3), with probability $\geq 1 - \delta$ we have

$$
\begin{aligned}
\ell_{\mathcal{D}}(g \circ f) - \ell_{\widehat{\mathcal{D}}^n}(g \circ f) &= \mathbb{E}_{(x,y) \sim \mathcal{D}}[h_{g,f}(x, y)] - \frac{1}{n} \sum_{i=1}^{n} h_{g,f}(x_i, y_i) \\
&\leq 2\text{Rad}_{\widehat{\mathcal{D}}^n}(\mathcal{L}_{\mathcal{G}, \mathcal{F}}) + 3c\sqrt{\frac{\ln(2/\delta)}{2n}},
\end{aligned} \tag{10}
$$

where the inequality is by the well-known Rademacher complexity uniform bound. Similarly,

$$
\begin{aligned}
\ell_{\widehat{\mathcal{D}}^n}(g \circ f) - \ell_{\mathcal{D}}(g \circ f) &= \mathbb{E}_{(x,y) \sim \mathcal{D}}[-h_{g,f}(x, y)] - \frac{1}{n} \sum_{i=1}^{n} -h_{g,f}(x_i, y_i) \\
&\leq 2\text{Rad}_{\widehat{\mathcal{D}}^n}(-\mathcal{L}_{\mathcal{G}, \mathcal{F}}) + 3c\sqrt{\frac{\ln(2/\delta)}{2n}} \\
&= 2\text{Rad}_{\widehat{\mathcal{D}}^n}(\mathcal{L}_{\mathcal{G}, \mathcal{F}}) + 3c\sqrt{\frac{\ln(2/\delta)}{2n}}.
\end{aligned} \tag{11}
$$

The probability that both events equation 10 and equation 11 happen can be upper bounded by union bound, i.e.,

$$\Pr((10) \wedge (11)) = 1 - \Pr((10)^c \vee (11)^c) \geq 1 - (\Pr((10)^c) + \Pr((11)^c)) \geq 1 - 2\delta.$$

Therefore, combining the above with probability $\geq 1 - \delta$ we have

$$|\ell_{\mathcal{D}}(g \circ f) - \ell_{\widehat{\mathcal{D}}^n}(g \circ f)| \leq 2\mathrm{Rad}_{\widehat{\mathcal{D}}^n}(\mathcal{L}_{\mathcal{G},\mathcal{F}}) + 3c\sqrt{\frac{\ln(4/\delta)}{2n}}. \tag{12}$$

With equation 12, we can prove the lemma as the following. Given $\forall \epsilon > 0$, by the definition of infimum there exists a $g_\epsilon \in \mathcal{G}$ such that

$$\ell_{\mathcal{D}}(g_\epsilon \circ f) < \inf_{g \in \mathcal{G}} \ell_{\mathcal{D}}(g \circ f) + \epsilon.$$

By equation 12, with probability $\geq 1 - \delta$ we have

$$\ell_{\widehat{\mathcal{D}}^n}(g_\epsilon \circ f) \leq \ell_{\mathcal{D}}(g_\epsilon \circ f) + 2\mathrm{Rad}_{\widehat{\mathcal{D}}^n}(\mathcal{L}_{\mathcal{G},\mathcal{F}}) + 3c\sqrt{\frac{\ln(4/\delta)}{2n}}.$$

Moreover, by definition

$$\inf_{g \in \mathcal{G}} \ell_{\widehat{\mathcal{D}}^n}(g \circ f) \leq \ell_{\widehat{\mathcal{D}}^n}(g_\epsilon \circ f).$$

Combining the above three inequalities we have

$$\inf_{g \in \mathcal{G}} \ell_{\widehat{\mathcal{D}}^n}(g \circ f) < \inf_{g \in \mathcal{G}} \ell_{\mathcal{D}}(g \circ f) + \epsilon + 2\mathrm{Rad}_{\widehat{\mathcal{D}}^n}(\mathcal{L}_{\mathcal{G},\mathcal{F}}) + 3c\sqrt{\frac{\ln(4/\delta)}{2n}}.$$

Letting $\epsilon \to 0$, we can see that

$$\inf_{g \in \mathcal{G}} \ell_{\widehat{\mathcal{D}}^n}(g \circ f) \leq \inf_{g \in \mathcal{G}} \ell_{\mathcal{D}}(g \circ f) + 2\mathrm{Rad}_{\widehat{\mathcal{D}}^n}(\mathcal{L}_{\mathcal{G},\mathcal{F}}) + 3c\sqrt{\frac{\ln(4/\delta)}{2n}}.$$

Similarly, we can derive the above inequality again but with $\mathcal{D}$ and $\widehat{\mathcal{D}}^n$ switched. Therefore,

$$|\inf_{g \in \mathcal{G}} \ell_{\widehat{\mathcal{D}}^n}(g \circ f) - \inf_{g \in \mathcal{G}} \ell_{\mathcal{D}}(g \circ f)| \leq 2\mathrm{Rad}_{\widehat{\mathcal{D}}^n}(\mathcal{L}_{\mathcal{G},\mathcal{F}}) + 3c\sqrt{\frac{\ln(4/\delta)}{2n}}.$$

Since the above inequality holds for $\forall f \in \mathcal{F}$, taking the supremum over $f \in \mathcal{F}$ gives the lemma. $\square$

**Lemma A.3.** *Assuming the individual loss function $\ell : \mathcal{Y} \times \mathcal{Y} \to [0, c]$, given any distributions $\mathcal{D}_S, \mathcal{D}_T \in \mathcal{P}_{\mathcal{X} \times \mathcal{Y}}$ and $\forall \delta > 0$, with probability $\geq 1 - \delta$ we have*

$$d_{\mathcal{F}_A}(\mathcal{D}_S, \mathcal{D}_T) \leq d_{\mathcal{F}_A}(\widehat{\mathcal{D}}_S^n, \widehat{\mathcal{D}}_T^n) + 2(\mathrm{Rad}_{\widehat{\mathcal{D}}_S^n}(\mathcal{L}_{\mathcal{G},\mathcal{F}_A}) + \mathrm{Rad}_{\widehat{\mathcal{D}}_T^n}(\mathcal{L}_{\mathcal{G},\mathcal{F}_A})) + 6c\sqrt{\frac{\ln(8/\delta)}{2n}}.$$

*Proof.* By Proposition 3.2, we apply the triangle inequality to derive

$$d_{\mathcal{F}_A}(\mathcal{D}_S, \mathcal{D}_T) \leq d_{\mathcal{F}_A}(\mathcal{D}_S, \widehat{\mathcal{D}}_T^n) + d_{\mathcal{F}_A}(\widehat{\mathcal{D}}_T^n, \mathcal{D}_T)$$
$$\leq d_{\mathcal{F}_A}(\widehat{\mathcal{D}}_S^n, \widehat{\mathcal{D}}_T^n) + d_{\mathcal{F}_A}(\widehat{\mathcal{D}}_T^n, \mathcal{D}_T) + d_{\mathcal{F}_A}(\widehat{\mathcal{D}}_S^n, \mathcal{D}_S).$$

By Lemma 3.1, we can apply the union bound argument (e.g., see the proof of Lemma 3.1) to bound $d_{\mathcal{F}_A}(\widehat{\mathcal{D}}_T^n, \mathcal{D}_T)$ and $d_{\mathcal{F}_A}(\widehat{\mathcal{D}}_S^n, \mathcal{D}_S)$. That being said, $\forall \delta' > 0$ with probability $\geq 1 - 2\delta'$ we have

$$d_{\mathcal{F}_A}(\widehat{\mathcal{D}}_S^n, \mathcal{D}_S) \leq 2\mathrm{Rad}_{\widehat{\mathcal{D}}_S^n}(\mathcal{L}_{\mathcal{G},\mathcal{F}_A}) + 3c\sqrt{\frac{\ln(4/\delta')}{2n}}$$

$$d_{\mathcal{F}_A}(\widehat{\mathcal{D}}_T^n, \mathcal{D}_T) \leq 2\mathrm{Rad}_{\widehat{\mathcal{D}}_T^n}(\mathcal{L}_{\mathcal{G},\mathcal{F}_A}) + 3c\sqrt{\frac{\ln(4/\delta')}{2n}}.$$

Therefore,

$$d_{\mathcal{F}_A}(\widehat{\mathcal{D}}_T^n, \mathcal{D}_T) + d_{\mathcal{F}_A}(\widehat{\mathcal{D}}_S^n, \mathcal{D}_S) \leq 2(\mathrm{Rad}_{\widehat{\mathcal{D}}_S^n}(\mathcal{L}_{\mathcal{G},\mathcal{F}_A}) + \mathrm{Rad}_{\widehat{\mathcal{D}}_T^n}(\mathcal{L}_{\mathcal{G},\mathcal{F}_A})) + 6c\sqrt{\frac{\ln(4/\delta')}{2n}}.$$

Denoting $\delta = 2\delta'$ gives the lemma. $\square$

**Theorem A.3** (Theorem 3.3 Restated). *Given $\forall \mathcal{D}_S, \mathcal{D}_T \in \mathcal{P}_{\mathcal{X} \times \mathcal{Y}}$, for $\forall \delta > 0$ with probability $\geq 1 - \delta$ we have*

$$\tau(A; \widehat{\mathcal{D}}_S^n, \mathcal{D}_T) \leq d_{\mathcal{F}_A}(\widehat{\mathcal{D}}_S^n, \widehat{\mathcal{D}}_T^n) + 2\text{Rad}_{\widehat{\mathcal{D}}_T^n}(\mathcal{L}_{\mathcal{G}, \mathcal{F}_A}) + 4\text{Rad}_{\widehat{\mathcal{D}}_S^n}(\mathcal{L}_{\mathcal{G}, \mathcal{F}_A}) + 9c\sqrt{\frac{\ln(8/\delta)}{2n}}.$$

*Proof.* For $\forall \delta > 0$, from the proof of Lemma A.3 we can see that with probability $\geq 1 - \delta$:

$$d_{\mathcal{F}_A}(\widehat{\mathcal{D}}_S^n, \mathcal{D}_S) \leq 2\text{Rad}_{\widehat{\mathcal{D}}_S^n}(\mathcal{L}_{\mathcal{G}, \mathcal{F}_A}) + 3c\sqrt{\frac{\ln(8/\delta)}{2n}} \qquad (13)$$

$$d_{\mathcal{F}_A}(\widehat{\mathcal{D}}_T^n, \mathcal{D}_T) \leq 2\text{Rad}_{\widehat{\mathcal{D}}_T^n}(\mathcal{L}_{\mathcal{G}, \mathcal{F}_A}) + 3c\sqrt{\frac{\ln(8/\delta)}{2n}},$$

and Lemma A.3 holds. Therefore

$$\tau(A; \widehat{\mathcal{D}}_S^n, \mathcal{D}_T) = \inf_{g \in \mathcal{G}} \ell_{\mathcal{D}_T}(g \circ f_A^{\widehat{\mathcal{D}}_S^n}) - \ell_{\widehat{\mathcal{D}}_S^n}(g_A^{\widehat{\mathcal{D}}_S^n} \circ f_A^{\widehat{\mathcal{D}}_S^n})$$

$$\leq \inf_{g \in \mathcal{G}} \ell_{\mathcal{D}_T}(g \circ f_A^{\widehat{\mathcal{D}}_S^n}) - \inf_{g \in \mathcal{G}} \ell_{\widehat{\mathcal{D}}_S^n}(g \circ f_A^{\widehat{\mathcal{D}}_S^n})$$

$$= \inf_{g \in \mathcal{G}} \ell_{\mathcal{D}_T}(g \circ f_A^{\widehat{\mathcal{D}}_S^n}) - \inf_{g \in \mathcal{G}} \ell_{\mathcal{D}_S}(g \circ f_A^{\widehat{\mathcal{D}}_S^n}) + \inf_{g \in \mathcal{G}} \ell_{\mathcal{D}_S}(g \circ f_A^{\widehat{\mathcal{D}}_S^n}) - \inf_{g \in \mathcal{G}} \ell_{\widehat{\mathcal{D}}_S^n}(g \circ f_A^{\widehat{\mathcal{D}}_S^n})$$

$$\leq d_{\mathcal{F}_A}(\mathcal{D}_S, \mathcal{D}_T) + d_{\mathcal{F}_A}(\widehat{\mathcal{D}}_S^n, \mathcal{D}_S)$$

$$\leq d_{\mathcal{F}_A}(\mathcal{D}_S, \mathcal{D}_T) + 2\text{Rad}_{\widehat{\mathcal{D}}_S^n}(\mathcal{L}_{\mathcal{G}, \mathcal{F}_A}) + 3c\sqrt{\frac{\ln(8/\delta)}{2n}}$$

$$\leq d_{\mathcal{F}_A}(\widehat{\mathcal{D}}_S^n, \widehat{\mathcal{D}}_T^n) + 2\text{Rad}_{\widehat{\mathcal{D}}_T^n}(\mathcal{L}_{\mathcal{G}, \mathcal{F}_A}) + 4\text{Rad}_{\widehat{\mathcal{D}}_S^n}(\mathcal{L}_{\mathcal{G}, \mathcal{F}_A}) + 9c\sqrt{\frac{\ln(8/\delta)}{2n}},$$

where the first inequality is by definition of infimum, the second inequality is by the Definition 2, the third inequality is by equation 13 and the last inequality is by Lemma A.3. □

## B  DATA AUGMENTATION (DA) AS REGULARIZATION

In this section, we discuss data augmentation (DA) as a concrete example of regularization for training feature extractor $f_A^{\mathcal{D}_S}$, and explore its impact on the function class $\mathcal{F}_A$ discussed in Section 3.

Empirical research has shown evidence of the regularization effect of DA (Hernández-García & König, 2018a;b). However, there is a lack of theoretical analysis, and thus we aim to construct a theoretical framework to understand under what sufficient conditions DA can be viewed as regularization on the feature extractor function class $\mathcal{F}_A$. We categorize DA into *feature-level DA* and *data-level DA*, and for each category, we analyze different DA algorithms to characterize the sufficient conditions under which DA regularizes the function class $\mathcal{F}_A$. Combined with analysis in Theorem 3.3, we also provide concrete sufficient conditions to tighten the upper bound of relative transferability $\tau(A; \widehat{\mathcal{D}}^n, \mathcal{D}_T)$.

**General Settings.** For the following discussion we apply a general DA setting of affine transformation (Perez & Wang, 2017), taking the form of $x_\star = W_\star^\top x + b_\star$, where $(x, x_\star)$ is a pair of the original and augmented samples, $(W_\star, b_\star)$ are parameters representing specific DA policies. We set $g : \mathbb{R}^d \to \mathbb{R}$ as the linear layer corresponding to the weight matrix $W_g$, which will be composed with the feature extractor $f : \mathbb{R}^m \to \mathbb{R}^d$. We use squared loss for $\ell : \mathbb{R} \times \mathbb{R} \to \mathbb{R}$, and let $\ell_{\widehat{\mathcal{D}}^n, A}(g \circ f)$ be the objective function given by training algorithm $A$ from Theorem 3.3.

### B.1  FEATURE-LEVEL DA ($A^{FL}$)

Feature-level DA (Wong et al., 2016; DeVries & Taylor, 2017) requires the transformation to be performed in the learned feature space, which gives us an augmented feature $W_\star f(x) + b_\star$. We use Loss-Averaging algorithm where we take an average of the loss over augmented features for

training. Denote the training algorithm based on feature-level DA as $A^{FL}$, the objective function is as below.

$$\ell_{\widehat{\mathcal{D}}^n, A^{FL}}(g \circ f) = \frac{1}{n} \sum_{i=1}^{n} \mathbb{E}_{W_\star, b_\star} \ell\Big(g \circ \big(W_\star f(x_i) + b_\star\big), y_i\Big).$$

**Theorem B.1.** *Apply feature-level DA with affine transformation parameters $(W_\star, b_\star)$ s.t.   1) $\mathbb{E}_{W_\star}[W_\star] = \mathbb{I}_m$; 2) $W_\star \not\equiv \mathbb{I}_m$ (i.e., $W_\star$ is not an identity matrix); 3) $\mathbb{E}_{b_\star}[b_\star] = \vec{0}_m$; 4) $W_\star$ and $b_\star$ are independent. Set $\ell : \mathbb{R} \times \mathbb{R} \to \mathbb{R}$ as squared loss; Define $\Delta_{W_\star} := W_\star - \mathbb{I}_m$, then we have*

$$\ell_{\widehat{\mathcal{D}}^n, A^{FL}}(g \circ f) = \ell_{\widehat{\mathcal{D}}^n, A}(g \circ f) + \Omega_{A^{FL}},$$

*where $\Omega_{A^{FL}} = \frac{1}{n} \sum_{i=1}^{n} \mathbb{E}_{W_\star} \Big[ \big\| f(x_i)^\top \Delta_{W_\star} W_g \big\|_2^2 \Big] + \mathbb{E}_{b_\star} \Big[ \big\| b_\star^\top W_g \big\|_2^2 \Big].$*

*Proof.* $\ell''(W_g^\top \circ f(x_i)) = 2$ for $\ell$ as squared loss. Apply Taylor expansion to $\ell\Big(g \circ \big(W_\star f(x_i) + b_\star\big), y_i\Big)$ around $f(x_i)$, all higher-than-two order terms will vanish:

$$\mathbb{E}_{W_\star, b_\star} \Big[ \ell\Big(g \circ \big(W_\star f(x_i) + b_\star\big), y_i\Big) \Big]$$

$$= \mathbb{E}_{W_\star, b_\star} \Big[ \ell\Big(W_g^\top \circ f(x_i), y_i\Big) + W_g^\top(\Delta_{W_\star} f(x_i) + b_\star) \ell'(W_g^\top \circ f(x_i), y_i) +$$

$$\frac{1}{2} W_g^\top (\Delta_{W_\star} f(x_i) + b_\star)(\Delta_{W_\star} f(x_i) + b_\star)^\top \ell''(W_g^\top \circ f(x_i), y_i) W_g \Big]$$

$$= \ell\Big(W_g^\top \circ f(x_i), y_i\Big) + \mathbb{E}_{W_\star, b_\star} \Big[ W_g^\top (\Delta_{W_\star} f(x_i) + b_\star)(\Delta_{W_\star} f(x_i) + b_\star)^\top W_g \Big]$$

$$= \ell\Big(W_g^\top \circ f(x_i), y_i\Big) + \mathbb{E}_{W_\star} \Big[ \big\| f(x_i)^\top \Delta_{W_\star} W_g \big\|_2^2 \Big] + \mathbb{E}_{b_\star} \Big[ \big\| b_\star^\top W_g \big\|_2^2 \Big];$$

The second equality holds since $\mathbb{E}_{\Delta_{W_\star}} = \mathbb{E}_{W_\star}[W_\star - \mathbb{I}_m] = 0_{(m,m)}$ and $\mathbb{E}_{b_\star} = \vec{0}_m$; The third equality holds since $W_i$ and $b_i$ are independent. Therefore, $\ell_{\widehat{\mathcal{D}}^n, A^{FL}}(g \circ f) := \frac{1}{n} \sum_{i=1}^{n} \Big[ \mathbb{E}_{W_\star, b_\star} \ell\Big(g \circ \big(W_\star f(x_i) + b_\star\big), y_i\Big) \Big] = \ell_{\widehat{\mathcal{D}}^n, A}(g \circ f) + \Omega_{A^{FL}}.$ $\qquad \square$

***Interpretation***. $\Omega_{A^{FL}}$ is composed of two segments: 1) $l_2$ regularization to an $f$-dependent scalar averaged over $W_\star$ and $x_i$; 2) $l_2$ regularization to an $f$-independent scalar averaged over $b_\star$. Due to the regularization effect on $f$ from the first segment of $\Omega_{A^{FL}}$, we can reasonably expect the function class $\mathcal{F}_{A'}$ enabled by $A^{FL}$ to be a subset of that enabled by a general training algorithm $A$.

***Sufficient conditions***. Combined with Theorem 3.3, the *sufficient conditions* to tighten the upper bound $d_{\mathcal{F}_A}(\widehat{\mathcal{D}}_S^n, \widehat{\mathcal{D}}_T^n)$ for the relative transferability $\tau(A; \widehat{\mathcal{D}}_S, \mathcal{D}_T)$ are: feature-level DA ($A^{FL}$) with parameters satisfying: 1) $\mathbb{E}_{W_\star}[W_\star] = \mathbb{I}_m$; 2) $W_\star \not\equiv \mathbb{I}_m$; 3) $\mathbb{E}_{b_\star}[b_\star] = \vec{0}_m$; 4) $W_\star$ and $b_\star$ are independent.

### B.2 DATA-LEVEL DA ($A^{DL}$)

Data-level DA requires that the transformation to be performed in the input space to generate augmented samples $W_\star x + b_\star$. We cover analysis on two ubiquitous algorithms for data-level DA training: *Prediction-Averaging ($A_P^{DL}$)* (Lyle et al., 2019) and *Loss-Averaging ($A_L^{DL}$)* (Wong et al., 2016). The difference between $A_P^{DL}$ and $A_L^{DL}$ lies in whether we take the average of the prediction or the losses:

$$\ell_{\widehat{\mathcal{D}}^n, A_P^{DL}}(g \circ f) := \frac{1}{n} \sum_{i=1}^{n} \ell\big( \mathbb{E}_{W_\star, b_\star} [g \circ f(W_\star x_i + b_\star)], y_i \big); \tag{14}$$

$$\ell_{\widehat{\mathcal{D}}^n, A_L^{DL}}(g \circ f) := \frac{1}{n} \sum_{i=1}^{n} \mathbb{E}_{W_\star, b_\star} \big[ \ell\big(g \circ f(W_\star x_i + b_\star), y_i \big) \big].$$

**Theorem B.2.** *Define the data-level deviation caused by data-level DA $A^{DL} \in \{A^{DL}_P, A^{DL}_L\}$ with parameters $(W_\star, b_\star)$ from the original data sample as $\Delta_{x_i} := (W_\star - \mathbb{I}_m)x_i + b_\star$, and define $\Delta^3_x := \mathbb{E}_{x_i, W_\star, b_\star}\left[\left\|\Delta_{x_i}\right\|^3_2\right]$. Suppose we apply data-level DA s.t. 1) $\mathbb{E}_{W_\star}[W_\star] = \mathbb{I}_m$; 2) $\mathbb{E}_{b_\star}[b_\star] = \vec{0}_m$; 3) $\mathcal{O}(\Delta^j_x) \approx 0, \forall j \in \mathbb{N}_+, j \geq 3$; 4) $W_\star$ and $b_\star$ are independent. Define $\Delta_{W_\star} := W_\star - \mathbb{I}_m \in \mathbb{R}^{m \times m}$, $\Delta_{\widehat{y}_i} := W^\top_g f(x_i) - y_i \in \mathbb{R}$. Let $W^{(k)}_g \in \mathbb{R}$ be the $k^{th}$ dimension component of $W_g$ and then define $w_{i,(k)} := W^{(k)}_g \Delta_{\widehat{y}_i} \in \mathbb{R}$; Denote the Hessian matrix of the $k^{th}$ dimension component in $f(x_i)$ as $\mathcal{H}^{(k),i}_f$; Let $\nabla f$ be the Jacobian matrix of $f$, then we have*

$$\ell_{\widehat{\mathcal{D}}^n, A^{DL}}(g \circ f) = \ell_{\widehat{\mathcal{D}}^n, A}(g \circ f) + \Omega_{A^{DL}} + \mathcal{O}(\Delta^3_x),$$

*where $\Omega_{A^{DL}_P} = \frac{1}{n}\sum_{i=1}^n \sum_{k=1}^d w_{i,(k)}\left[tr\left(\mathbb{E}_{W_\star}[\Delta_{x_i}\Delta^\top_{x_i}]\mathcal{H}^{(k),i}_f\right)\right]$, where $\Delta_{x_i} = (W_\star - \mathbb{I})^\top x_i + b_\star$; $\Omega_{A^{DL}_L} = \Omega_{A^{DL}_P} + \frac{1}{n}\sum_{i=1}^n\left[\mathbb{E}_{W_\star}\left\|x^\top_i \Delta_{W_\star} \nabla f(x_i)W_g\right\|^2_2 + \mathbb{E}_{b_\star}\left\|b^\top_\star \nabla f(x_i)W_g\right\|^2_2\right]$.*

*Proof.* Let $\Delta_{f_i, A^{DL}_P} := \mathbb{E}_{W_\star, b_\star} f(W^\top_\star x_i + b_\star) - f(x_i)$, then

$$\begin{aligned}
\Delta_{f_i, A^{DL}_P} :=&\mathbb{E}_{W_\star, b_\star} f(W^\top_\star x_i + b_\star) - f(x_i)\\
=&\mathbb{E}_{W_\star, b_\star}\left[\nabla f(x_i)^\top(\Delta_{x_i})\right] + \frac{1}{2}\mathbb{E}_{W_\star, b_\star}\left[\Delta^\top_{x_i}\mathcal{H}^{(k),i}_f(x_i)\Delta_{x_i}\right]_d + \mathcal{O}(\mathbb{E}_{W_\star, b_\star}\|\Delta_{x_i}\|^3_2)\\
=&\frac{1}{2}\mathbb{E}_{W_\star, b_\star}\left[\Delta^\top_{x_i}\mathcal{H}^{(k),i}_f(x_i)\Delta_{x_i}\right]_d + \mathcal{O}(\mathbb{E}_{W_\star, b_\star}\|\Delta_{x_i}\|^3_2),
\end{aligned} \tag{15}$$

where $[\cdot^{(k)}]_d$ denotes a d-dimensional vector and $k$ denotes the $k^{th}$ dimension element. Since $\ell$ is squared loss, the third-and-higher derivative are 0, therefore, the third-and-higher order terms in Taylor expansion to $\ell\left(\mathbb{E}_{W_\star, b_\star}\left[g \circ f(W_\star x_i + b_\star)\right], y_i\right)$ around $f(x_i)$ will vanish:

$$\begin{aligned}
&\ell\left(\mathbb{E}_{W_\star, b_\star}\left[g \circ f(W_\star x_i + b_\star)\right], y_i\right)\\
=&\ell\left(g \circ f(x_i), y_i\right) + W^\top_g(\Delta_{f_i, A^{DL}_P})\ell'\left(g \circ f(x_i), y_i\right)+\\
&\frac{1}{2}W^\top_g(\Delta_{f_i, A^{DL}_P})(\Delta_{f_i, A^{DL}_P})^\top W_g \ell''\left(g \circ f(x_i), y_i\right)\\
=&\ell\left(g \circ f(x_i), y_i\right) + W^\top_g(\Delta_{f_i, A^{DL}_P})\ell'\left(g \circ f(x_i), y_i\right) + \mathcal{O}(\mathbb{E}_{W_\star, b_\star}\|\Delta_{x_i}\|^4_2)
\end{aligned} \tag{16}$$

Substitute Eq. (15) into the first-order term in Eq. (16), we have

$$\begin{aligned}
W^\top_g(\Delta_{f_i, A^{DL}_P})\ell'\left(g \circ f(x_i), y_i\right) =&W^\top_g \mathbb{E}_{W_\star, b_\star}\left[\Delta^\top_{x_i}\mathcal{H}^{(k),i}_f\Delta_{x_i}\right]_d \Delta_{\widehat{y}_i} + \mathcal{O}(\mathbb{E}_{W_\star, b_\star}\|\Delta_{x_i}\|^3_2)\\
=&\Delta_{\widehat{y}_i}\sum_{k=1}^d W^{(k)}_g \mathbb{E}_{W_\star, b_\star}\left[\Delta^\top_{x_i}\mathcal{H}^{(k),i}_f\Delta_{x_i}\right] + \mathcal{O}(\mathbb{E}_{W_\star, b_\star}\|\Delta_{x_i}\|^3_2)\\
=&\sum_{k=1}^d w_{i,(k)}tr\left(\mathbb{E}_{W_\star, b_\star}[\Delta_{x_i}\Delta^\top_{x_i}]\mathcal{H}^{(k),i}_f\right) + \mathcal{O}(\mathbb{E}_{W_\star, b_\star}\|\Delta_{x_i}\|^3_2).
\end{aligned} \tag{17}$$

Substitute Eq. (17) into Eq. (16), we have

$$\begin{aligned}
\ell\left(\mathbb{E}_{W_\star, b_\star}\left[g \circ f(W_\star x_i + b_\star)\right], y_i\right) =&\ell\left(g \circ f(x_i), y_i\right) + \sum_{k=1}^d w_{i,(k)}tr\left(\mathbb{E}_{W_\star, b_\star}[\Delta_{x_i}\Delta^\top_{x_i}]\mathcal{H}^{(k),i}_f\right)+\\
&\mathcal{O}(\mathbb{E}_{W_\star, b_\star}\|\Delta_{x_i}\|^3_2).
\end{aligned} \tag{18}$$

Substitute Eq. (18) into Eq. (14) which is the definition of $\ell_{\widehat{\mathcal{D}}^n, A^{DL}_P}(g \circ f)$, and recall that $\Delta^3_x := \mathbb{E}_{x_i, W_\star, b_\star}\left[\left\|\Delta_{x_i}\right\|^3_2\right]$, we have

$$\ell_{\widehat{\mathcal{D}}^n, A^{DL}_P}(g \circ f) := \frac{1}{n}\sum_{i=1}^n \ell\left(\mathbb{E}_{W_\star, b_\star}\left[g \circ f(W_\star x_i + b_\star)\right], y_i\right) = \ell_{\widehat{\mathcal{D}}^n, A}(g \circ f) + \Omega_{A^{DL}_P} + \mathcal{O}(\Delta^3_x).$$

$$\tag{19}$$

Let $\Delta_{f_i, A_L^{DL}} := f(W_\star^\top x_i + b_\star) - f(x_i) = \nabla f(x_i)^\top (\Delta_{W_\star} x_i + b_\star) + \mathcal{O}(\|\Delta_{x_i}\|_2^2)$.

Applying Taylor expansion to $\mathbb{E}_{W_\star, b_\star}\big[\ell\big(g \circ f(W_\star x_i + b_\star), y_i\big)\big]$ around $f(x_i)$ will give us

$$\mathbb{E}_{W_\star, b_\star}\big[\ell\big(g \circ f(W_\star x_i + b_\star), y_i\big)\big] = \ell\big(g \circ f(x_i), y_i\big) + W_g^\top \mathbb{E}_{W_\star, b_\star}\big[\Delta_{f_i, A_L^{DL}}\big]\ell'\big(g \circ f(x_i), y_i\big) +$$
$$\frac{1}{2} W_g^\top \mathbb{E}_{W_\star, b_\star}\big[(\Delta_{f_i, A_L^{DL}})(\Delta_{f_i, A_L^{DL}})^\top\big] W_g \ell''\big(g \circ f(x_i), y_i\big) \tag{20}$$

Since $\mathbb{E}_{W_\star, b_\star} \Delta_{f_i, A_L^{DL}} = \Delta_{f_i, A_P^{DL}}$, the first-order term in Eq. (20) is exactly Eq. (17):

$$W_g^\top \mathbb{E}_{W_\star, b_\star}\big[\Delta_{f_i, A_L^{DL}}\big]\ell'\big(g \circ f(x_i), y_i\big)$$
$$= W_g^\top \Delta_{f_i, A_P^{DL}} \ell'\big(g \circ f(x_i), y_i\big)$$
$$= \sum_{k=1}^d w_{i,(k)} tr\big(\mathbb{E}_{W_\star, b_\star}[\Delta_{x_i} \Delta_{x_i}^\top] \mathcal{H}_f^{(k),i}\big) + \mathcal{O}(\mathbb{E}_{W_\star, b_\star} \|\Delta_{x_i}\|_2^3) \tag{21}$$

The second-order term in Eq. (20) is

$$\frac{1}{2} W_g^\top \mathbb{E}_{W_\star, b_\star}\big[(\Delta_{f_i, A_L^{DL}})(\Delta_{f_i, A_L^{DL}})^\top\big] W_g \ell''\big(g \circ f(x_i), y_i\big)$$
$$= W_g^\top \mathbb{E}_{W_\star, b_\star}\big[(\nabla f(x_i)^\top (\Delta_{W_\star} x_i + b_\star)(\Delta_{W_\star} x_i + b_\star)^\top \nabla f(x_i)\big] W_g + \mathcal{O}(\mathbb{E}_{W_\star, b_\star} \|\Delta_{x_i}\|_2^4)$$
$$= \mathbb{E}_{W_\star} \big\|x_i^\top \Delta_{W_\star}^\top \nabla f(x_i) W_g\big\|_2^2 + \mathbb{E}_{b_\star} \big\|b_\star^\top \nabla f(x_i) W_g\big\|_2^2 + \mathcal{O}(\mathbb{E}_{W_\star, b_\star} \|\Delta_{x_i}\|_2^4) \tag{22}$$

Substituting Eq. (21) and Eq. (22) into Eq. (20), we have

$$\mathbb{E}_{W_\star, b_\star}\big[\ell\big(g \circ f(W_\star x_i + b_\star), y_i\big)\big]$$
$$= \ell\big(g \circ f(x_i), y_i\big) + \sum_{k=1}^d w_{i,(k)} tr\big(\mathbb{E}_{W_\star, b_\star}[\Delta_{x_i} \Delta_{x_i}^\top] \mathcal{H}_f^{(k),i}\big) +$$
$$\mathbb{E}_{W_\star} \big\|x_i^\top \Delta_{W_\star} \nabla f(x_i) W_g\big\|_2^2 + \mathbb{E}_{b_\star} \big\|b_\star^\top \nabla f(x_i) W_g\big\|_2^2 + \mathcal{O}(\mathbb{E}_{W_\star, b_\star} \|\Delta_{x_i}\|_2^4) \tag{23}$$

Substitute Eq. (23) into the definition of $\ell_{\widehat{\mathcal{D}}^n, A_L^{DL}}(g \circ f)$, then

$$\ell_{\widehat{\mathcal{D}}^n, A_L^{DL}}(g \circ f) := \frac{1}{n} \sum_{i=1}^n \mathbb{E}_{W_\star, b_\star}\big[\ell\big(g \circ f(W_\star x_i + b_\star), y_i\big)\big] = \ell_{\widehat{\mathcal{D}}^n, A}(g \circ f) + \Omega_{A_L^{DL}} + \mathcal{O}(\Delta_x^3)$$
$$\tag{24}$$

The proof is complete by Eq. (19) and Eq. (24). $\qquad\square$

***Interpretation***. $\Omega_{A_P^{DL}}$ and $\Omega_{A_L^{DL}}$ turn out to be: 1) $\Omega_{A_P^{DL}}$ is a weighted trace expectation dependent on the Hessian matrix of $f$; 2) $\Omega_{A_L^{DL}}$ is equivalent to $\Omega_{A_P^{DL}}$ together with the summation of two norm expectations dependent on $\nabla f$. Therefore, the data-level DA algorithms $A_P^{DL}$ and $A_L^{DL}$ are expected to regularize $f$ so that the $f$ function class $\mathcal{F}_A^{DL}$ enabled by $A^{DL} \in \{A_P^{DL}, A_L^{DL}\}$ would be reasonably expected as a subset of $\mathcal{F}_A$ enabled by general training algorithm $A$.

***Sufficient conditions***. Combined with Theorem 3.3, the *sufficient conditions* indicated here to tighten the upper bound $d_{\mathcal{F}_A}(\widehat{\mathcal{D}}_S^n, \widehat{\mathcal{D}}_T^n)$ of the relative transferability $\tau(A; \widehat{\mathcal{D}}_S, \mathcal{D}_T)$ are: data-level DA ($A^{DL}$) with DA parameters satisfying that 1) $\mathbb{E}_{W_\star}[W_\star] = \mathbb{I}_m$; 2) $\mathbb{E}_{b_\star}[b_\star] = \vec{0}_m$; 3) $\mathcal{O}(\Delta_x^j) \approx 0, \forall j \in \mathbb{N}_+, j \geq 3$; 4) $W_\star$ and $b_\star$ are independent.

***Empirical verification***. We further provide empirical verification in Section 4 for the sufficient conditions above, investigating the concrete cases of DA methods: *1) Gaussian noise* satisfies the sufficient conditions, then we empirically show that Gaussian noise improves domain transferability while robustness decreases a bit (Figure 5); *2) Rotation*, which rotates input image with a predefined fixed angle with predefined fixed probability, violates $\mathbb{E}_{W_\star}[W_\star] = \mathbb{I}_m$, and we empirically show that rotation barely affect domain transferability (Figure 14 in Appendix D.3); *Translation*, which moves the input image for a predefined distance along a pre-selected axis with fixed probability, violates $\mathbb{E}_{b_\star}[b_\star] = \vec{0}_m$ (Figure 14 in Appendix D.3).

**Corollary B.2.1.** *If the neural network in Theorem B.2 is activated by Relu or Max-pooling, then Theorem B.2 becomes*

$$\ell_{\widehat{\mathcal{D}}^n, A^{DL}}(g \circ f) = \ell_{\widehat{\mathcal{D}}^n, A}(g \circ f) + \Omega_{A^{DL}} + \mathcal{O}(\Delta_x^3),$$

*where* $\Omega_{A_P^{DL}} = 0$; $\Omega_{A_L^{DL}} = \frac{1}{n} \sum_{i=1}^n \left[ \mathbb{E}_{W_\star} \left\| x_i^\top \Delta_{W_\star} \nabla f(x_i) W_g \right\|_2^2 + \mathbb{E}_{b_\star} \left\| b_\star^\top \nabla f(x_i) W_g \right\|_2^2 \right].$

*Proof.* Denote an $L-$layer NN $g \circ f(x) := W_g^\top \cdot z^{[L-1]}$, where $z^{[l]} := \sigma^{[l]}(W_{[l-1]}^\top \cdot z^{[l-1]})$, $l = 1, 2, 3, ..., L - 1$; Define that $\sigma^{[0]}(W_{[0]}^\top \cdot z^{[0]}) := x$, then $\nabla^2(g \circ f(x)) = 0$ (B.2 of Zhang et al. (2020)). Since $\nabla^2(g \circ f(x)) = W_g^\top \cdot \nabla^2 f(x)$, we have $\nabla^2 f(x) = 0$.

Combine this with Theorem B.2, we have

$$\Omega_{A_P^{DL}} = \frac{1}{n} \sum_{i=1}^n \sum_{k=1}^d w_{i,(k)} \left[ tr\left( \mathbb{E}_{W_\star}[\Delta_{x_i} \Delta_{x_i}^\top] \mathcal{H}_f^{(k),i} \right) \right] = 0;$$

$$\Omega_{A_L^{DL}} = \Omega_{A_P^{DL}} + \frac{1}{n} \sum_{i=1}^n \left[ \mathbb{E}_{W_\star} \left\| x_i^\top \Delta_{W_\star} \nabla f(x_i) W_g \right\|_2^2 + \mathbb{E}_{b_\star} \left\| b_\star^\top \nabla f(x_i) W_g \right\|_2^2 \right]$$

$$= \frac{1}{n} \sum_{i=1}^n \left[ \mathbb{E}_{W_\star} \left\| x_i^\top \Delta_{W_\star} \nabla f(x_i) W_g \right\|_2^2 + \mathbb{E}_{b_\star} \left\| b_\star^\top \nabla f(x_i) W_g \right\|_2^2 \right].$$

$\square$

***Remark.*** Corollary B.2.1 analyzes special cases (Relu/ Max-pooling activation) of Theorem B.2, giving notably different regularization effect: in these cases the $A_P^{DL}$ (average the prediction) fails as a regularizer, therefore, doesn't fulfill our sufficient conditions for improving domain transferability (Theorem 3.3); $A_L^{DL}$ (average the loss) only reserves the regularization on $\nabla f$-dependent norms, but no longer regularizes $\mathcal{H}_f(x)$. Since $A_L^{DL}$ still induces regularization, the induced sufficient conditions analyzed after Theorem B.2 for promoting domain transferability won't be affected.

## C  ADVERSARIAL TRAINING AS A REGULARIZER

In this section, we show, under certain conditions, why adversarial training may improve domain generalization by viewing adversarial training as a function class regularizer.

We first provide some notation. Let

$$\mathcal{F} = \{ f_\theta(\cdot) = W^L \phi^{L-1}(W^{L-1} \phi^{L-2}(\dots) + b^{L-1}) + b^L \}$$

where $\phi^j$ are activations, $W^j, b^j$ are weight matrix and bias vector, $\theta$ is the collection of parameters (i.e. $\theta = (W^1, b^1, \dots, W^L, b^L)$. For the rest of the article, assume that $\phi^j$ are just ReLUs.

Now fix $x \in \mathcal{X}$. Define the preactivation as

$$\widetilde{x}^1 := W^1 x + b^1$$
$$\widetilde{x}^j := W^j \phi^{j-1}(\widetilde{x}^{j-1}) + b^j , \, j \geq 2$$

Define the activation pattern $\phi_x := (\phi_x^1, \dots, \phi_x^{L-1}) \in \{0, 1\}^m$ such that for each $j \in [L-1]$

$$\phi_x^j = \mathbb{1}(\widetilde{x}^j > 0)$$

where $\mathbb{1}$ is applied elementwise.

Now, given an activation pattern $\phi \in \{0, 1\}^m$, we define the preimage $X(\phi) := \{x \in \mathbb{R}^d : \phi_x = \phi\}$

**Theorem C.1.** *(In the proof of theorem 1 in (Roth et al., 2020))*

*Let* $\epsilon > 0$ *s.t.* $B_\epsilon^p(x) \subset X(\phi_x)$ *where* $B_\epsilon^p(x)$ *denotes the* $l_p$ *ball centered at* $x$ *with radius* $\epsilon$. *Let* $p = \{1, 2, \infty\}$ *and* $q$ *be the Holder conjugate of* $p$ *(i.e.* $\frac{1}{p} + \frac{1}{q} = 1$*). Then*

$$\mathbb{E}_{(x,y) \sim P}[l(y, f(x)) + \lambda \max_{x^* \in B_\epsilon^p(x)} \|f(x) - f(x^*)\|_q] = \mathbb{E}_{(x,y) \sim P}[l(y, f(x)) + \lambda \cdot \epsilon \max_{v^* : \|v^*\|_p \leq 1} \|J_{f(x)} v\|_q]$$

**Interpretation:** This theorem provides an equivalence between the objective functions for adversarial training (left term) and jacobian regularization (right term). We give some intuition on the size of $\epsilon$. Let us first consider a shallow 2 layer network $f(x) = W^2\phi(W^1x+b^1)$. Suppose $W^2 \in \mathbb{R}^{m_2 \times m_1}$ and $W^1 \in \mathbb{R}^{m_1 \times d}$. Given a matrix M, let $M_j$ denote the $j$th row of $M$. We study the activation pattern $\phi_x$ which equals

$$\phi_x = (\phi_x^1) = \begin{pmatrix} \mathbb{1}\{W_1^1x + b_1^1\} \\ \vdots \\ \mathbb{1}\{W_{m_1}^1x + b_{m_1}^1\} \end{pmatrix}$$

We wish to compute the largest radius $\epsilon$ such that the activation pattern $\phi_x$ is constant within $B_\epsilon^2(x)$. This is simply the distance from $x$ to the closest hyperplane of the form $H_{W_j^1,b_j^1} = \{x \in \mathbb{R}^d : W_j^1x + b_j^1 = 0\}$ where $j = 0, \ldots, m_1$ (i.e. $\epsilon = \min_j \text{dist}(x, H_{W_j^1,b_j^1})$). In particular, if $W^1 = I_{d \times d}$ and $b^1 = \mathbf{0}$, $\epsilon = \min_{j \in d}|x_j|$.

Furthermore, we note that $\epsilon$ is nondecreasing as a function of the number of layers. However, it has been observed empirically in (Roth et al., 2020) that approximate correspondence holds in a much larger ball.

**Definition 4.** *(source and target function class) Let $\mathcal{G}^S, \mathcal{G}^T$ be fine tuning function classes for source and target domains, respectively. We define the class of source models as*

$$\mathcal{H}^S = \mathcal{G}^S \circ \mathcal{F} = \{g^S \circ f_\theta : g^S \in \mathcal{G}^S, f_\theta \in \mathcal{F}\}$$

*and the class of target models as*

$$\mathcal{H}^T = \mathcal{G}^T \circ \mathcal{F} = \{g^T \circ f_\theta : g^T \in \mathcal{G}^T, f_\theta \in \mathcal{F}\}$$

**Definition 5.** *(empirical training objective with jacobian regularization) Let $\lambda, \epsilon > 0$. Take any hypothesis $h_\theta = g^S \circ f_\theta \in \mathcal{H}^S$. Let $\hat{R}(h_\theta) = \frac{1}{n}\sum_{i=1}^n \ell(h_\theta(x_i), y_i)$ denote the empirical risk where $l(\hat{y}, y) = \|\hat{y} - y\|^2$. We define the empirical training objective with jacobian regularization as*

$$\text{Obj}_\lambda^A(h_\theta) = \hat{R}(h_\theta) + \frac{\lambda \cdot \epsilon}{n}\sum_{i=1}^n \|J_{h_\theta}(x_i)\|_2$$

**Theorem C.2.** *Fix regularization strength $\lambda > 0$. Define*

$$\mathcal{F}_\lambda^A = \{f_\theta^A \in \mathcal{F} : \exists g^S \in \mathcal{G}^S \text{ s.t. } \text{Obj}_\lambda^A(g^S \circ f_\theta^A) \leq \text{Obj}_\lambda^A(\mathbf{0})\}$$

*where $\mathbf{0}$ denotes the zero function (i.e. the class of feature extractors that outperform the zero function). Suppose $(x, y) \in \mathcal{X} \times \mathcal{Y}$ is bounded such that $\max(\|x\|_\infty, \|y\|_2) \leq R$. Fix $\delta > 0$. Suppose we additionally restrict our fine tuning class models to linear models where*

$$G^S = \{W : W \in \mathbb{R}^{d \times n}, n \geq 1, \min_j \|W_j\|_2 \geq \delta\}$$

*(where $W_j$ is the jth column of $W$) and*

$$G^T = \{W : W \in \mathbb{R}^{d \times n}, n \geq 1\}$$

*(Here we are abusing notation to let $g^S \in G^S$ to denote the last linear layer as well as the fine tuning function).*

*Then for $0 \leq \lambda_1 < \lambda_2$*

$$\mathcal{F}_{\lambda_2}^A \subsetneq \mathcal{F}_{\lambda_1}^A \subsetneq \mathcal{F}$$

*(where $\subsetneq$ denotes proper subset). In particular, if $H_\lambda^{A,T} = \mathcal{G}^T \circ \mathcal{F}_\lambda^A$, we have*

$$\mathcal{H}_{\lambda_2}^{A,T} \subsetneq \mathcal{H}_{\lambda_1}^{A,T} \subsetneq \mathcal{H}^T$$

**Interpretation:**

At the high level, this theorem captures the idea that minimizing the empirical risk with jacobian regularization puts a constraint on the set of feature extractors. In particular, $\mathcal{F}_{\lambda_1}^A$ represents the potential class of feature extractors we select after training with jacobian regularization. Therefore,

the class of fine tuned models $H_{\lambda_1}^{A,T}$ with feature extractors trained with jacobian regularization for the target domain is smaller than the class of fine tuned models $\mathcal{H}$ with feature extractors trained without any regularization. Furthermore, we show that the space of feature extractors shrinks as we increase the regularization stength $\lambda$. Since we showed in section 3.3 that smaller function classes have smaller $d_{\mathcal{F}_A}$, this theorem shows that jacobian regularization reduces $d_{\mathcal{F}_A}$. To connect back to adversarial training, if $\epsilon$ satisfies the hypothesis in theorem $C.1$, we have that

$$\mathbb{E}_{(x,y)\sim P}[l(y,f(x)) + \lambda \max_{x^* \in B_\epsilon^p(x)} \|f(x) - f(x^*)\|_q] = \mathbb{E}_{(x,y)\sim P}[l(y,f(x)) + \lambda \cdot \epsilon \max_{v^*:\|v^*\|_p \leq 1} \|J_{f(x)}v\|_q]$$

Therefore, minimizing the training objective with jacobian regularization is equivalent to minimizing the adversarial training objective. Using this connection, this theorem essentially shows that, given sufficient number of samples, adversarial training reduces the class of feature extractors which in turn reduces $d_{\mathcal{F}_A}$.

Finally, we comment on the assumption that $\|g^S\| > \delta$. Since $\delta > 0$ is arbitrary, we can make it as small as we like and thus we are essentially excluding the $\mathbf{0}$ last layer which is hardly a constraint on the function class. This assumption is necessary as we are considering regularization on the whole model $g \circ f$ as opposed to regularization on just the feature extractor. Thus, this assumption prevents the scenario where only the last linear layer is regularized.

*Proof.* We first show that if $0 \leq \lambda_1 < \lambda_2$, we have that
$$\mathcal{F}_{\lambda_2}^A \subsetneq \mathcal{F}_{\lambda_1}^A \subsetneq \mathcal{F}$$

We first prove the following lemma

**Lemma C.1.** *Suppose the conditions of theorem $C.2$ are satisfied. Suppose additionally we have that $\overline{y} := \frac{1}{n}\sum_{i=1}^d y_i \neq 0$ (note this occurs with probability 1 if marginal distribution over $\mathcal{Y}$ is continuous). Then for every $\lambda \geq 0$, there exists a function $f_\theta \in \mathcal{F}_\lambda^A$ and a fine tuning layer $g^* \in \mathcal{G}^S$ such that*

$$\text{Obj}_\lambda^A(g^* \circ f_\theta) = \inf_{g \in G^S} \text{Obj}_\lambda^A(g \circ f_\theta) = \text{Obj}_\lambda^A(\mathbf{0})$$

*Choose another $\lambda' \geq 0$ (can equal $\lambda$). Then there exists a $g^{*\prime} \in \mathcal{G}^S$ be the fine tuning layer such that $\inf_{g \in \mathcal{G}^S} \text{Obj}_{\lambda'}^A(g \circ f_\theta) = \text{Obj}_{\lambda'}^A(g^{*\prime} \circ f_\theta)$ and*

$$\frac{1}{n}\sum_{i=1}^n \|J_{g^{*\prime}\circ f_\theta}(x_i)\|_2 > 0$$

*Proof.* Fix $\alpha \geq 0$ and $c > \alpha \cdot R$. Set biases

$$b^1 = \begin{pmatrix} c \\ 0 \\ \vdots \\ 0 \end{pmatrix} \quad b^j = \mathbf{0}, \, j \geq 2$$

and weights

$$W^1 = \begin{pmatrix} \alpha & 0 & \dots & 0 \\ 0 & 0 & \dots & 0 \\ \vdots & \vdots & \vdots & \vdots \\ 0 & \dots & \dots & 0 \end{pmatrix} \quad W^j = \begin{pmatrix} 1 & 0 & \dots & 0 \\ 0 & 0 & \dots & 0 \\ \vdots & \vdots & \vdots & \vdots \\ 0 & \dots & \dots & 0 \end{pmatrix}, \, j \geq 2$$

Define $x_{i,j}$ be the $j$th entry of the data point $x_i$. Define

$$\alpha_i := \alpha \cdot x_{i,1}$$

$$\overline{\alpha} := \frac{1}{n}\sum_{i=1}^d \alpha_i$$

$$\overline{y} := \frac{1}{n}\sum_{i=1}^d y_i$$

Now we observe that for a fixed $\lambda \geq 0$ and any $g \in \mathcal{G}^S$, we have that

$$\text{Obj}_\lambda^A(g \circ f_\theta) = \hat{R}(g \circ f_\theta) + \lambda \cdot \epsilon \frac{1}{n} \sum_{i=1}^n \|J_{g \circ f_\theta}(x_i)\|_2$$

$$= \frac{1}{n} \sum_{i=1}^n \left\| (\alpha x_{i,1} + c) \begin{pmatrix} g_{11} \\ \vdots \\ g_{d1} \end{pmatrix} - y_i \right\|_2^2 + \lambda \cdot \epsilon \frac{1}{n} \sum_{i=1}^n \left\| g \begin{pmatrix} \alpha & 0 & \dots & 0 \\ 0 & 0 & \dots & 0 \\ \vdots & \vdots & \vdots & \vdots \\ 0 & \dots & \dots & 0 \end{pmatrix} \right\|_2$$

$$= \frac{1}{n} \sum_{i=1}^n \|(\alpha_i + c)g_1 - y_i\|_2^2 + \lambda \cdot \epsilon \frac{1}{n} \sum_{i=1}^n \alpha \|g_1\|_2$$

$$= \frac{1}{n} \sum_{i=1}^n \|(\alpha_i + c)g_1 - y_i\|_2^2 + \lambda \cdot \epsilon \alpha \|g_1\|_2$$

Therefore,

$$\inf_{g \in \mathcal{G}^S} \text{Obj}_\lambda^A(g \circ f_\theta)$$

is equivalent to solving

$$\inf_{w \in \mathbb{R}^d : \|w\|_2 \geq \delta} \frac{1}{n} \sum_{i=1}^n \|(\alpha_i + c)w - y_i\|_2^2 + \lambda \cdot \epsilon \alpha \|w\|_2 \tag{25}$$

Utilizing lagrange multipliers, we find the minimizer is

$$w = \delta \cdot \frac{\overline{y}}{\|\overline{y}\|} \tag{26}$$

when $c \geq \frac{\|\overline{y}\|_2}{\delta}$.

Now consider the function

$$S(c, \alpha) = \frac{1}{n} \sum_{i=1}^n \|(\alpha \cdot x_{i,1} + c)g_1 - y_i\|_2^2 + \lambda \cdot \epsilon \alpha \|g_1\|_2$$

Note that this function is continuous with respect to the input $(c, \alpha)$. Now fix $\alpha = 0, c = \frac{\|\overline{y}\|_2}{\delta}$. Set $w = \delta \cdot \frac{\overline{y}}{\|\overline{y}\|}$. Then we have that

$$S(\frac{\|\overline{y}\|_2}{\delta}, 0) = \frac{1}{n} \sum_{i=1}^n \|\overline{y} - y_i\|_2^2 < \frac{1}{n} \sum_{i=1}^n \|y_i\|_2^2 = \text{Obj}_\lambda^A(\mathbf{0})$$

The inequality comes from the fact that we assumed $\overline{y} \neq 0$ and noting that $\overline{y}$ is the minimizer of the function $p(z) = \frac{1}{n} \|z - y_i\|_2^2$. Continuity of $S$ ensures that there exists $\alpha_0 > 0$ such that

$$S(\frac{\|\overline{y}\|_2}{\delta}, \alpha_0) < \frac{1}{n} \sum_{i=1}^n \|y_i\|_2^2 = \text{Obj}_\lambda^A(\mathbf{0})$$

Now consider $U(t) = S((1+t)\frac{\|\overline{y}\|_2}{\delta}, (1+t)\alpha_0)$ for $t \geq 0$. Note that $U$ is continuous with respect to $t$. Furthermore, we note that $t \to \infty$ implies $U(t) \to \infty$ which implies there exists some time $t = T_f$ such that $U(T_f) > \text{Obj}_\lambda^A(\mathbf{0})$. Therefore, by the intermediate value theorem, there exists a time $t = T$ such that $U(T) = \text{Obj}_\lambda^A(\mathbf{0})$. Finally, set $c = (T+1)\frac{\|\overline{y}\|_2}{\delta}, \alpha = (T+1)\alpha_0$, and $g^*$ as the matrix where $g_1^* = \delta \cdot \frac{\overline{y}}{\|\overline{y}\|}$ and $\mathbf{0}$ for the other columns. By equation 25 and equation 26 we have

$$\text{Obj}_\lambda^A(g^* \circ f_\theta) = \inf_{g \in \mathcal{G}^S} \text{Obj}_\lambda^A(g \circ f_\theta) = U(T) = \text{Obj}_\lambda^A(\mathbf{0})$$

Furthermore, if we choose another $\lambda' \geq 0$, since $c = (T+1)\frac{\|\overline{y}\|_2}{\delta} > \frac{\|\overline{y}\|_2}{\delta}$ by equation 26, we have that

$$\text{Obj}_{\lambda'}^A(g^{*\prime} \circ f_\theta) = \inf_{g \in \mathcal{G}^S} \text{Obj}_{\lambda'}^A(g \circ f_\theta)$$

and

$$\frac{1}{n} \sum_{i=1}^{n} \left\| J_{g^{*\prime} \circ f_\theta}(x_i) \right\|_2 = \alpha \left\| g^{*\prime} \right\|_2 = \alpha \delta$$

which is nonzero as $\delta > 0$ and $\alpha = (T+1)\alpha_0 > 0$.

$\square$

Clearly, we have $\mathcal{F}_{\lambda_2}^A \subset \mathcal{F}_{\lambda_1}^A$. If we can show that $f_{\theta_1} \notin \mathcal{F}_{\lambda_2}^A$ then we have $\mathcal{F}_{\lambda_2}^A \subsetneq \mathcal{F}_{\lambda_1}^A$.

Using lemma $C.1$ we can find $f_{\theta_1} \in \mathcal{F}_{\lambda_1}^A$ such that

$$\inf_{g \in G^S} \mathrm{Obj}_{\lambda_1}^A (g \circ f_{\theta_1}) = \mathrm{Obj}_{\lambda_1}^A(\mathbf{0})$$

In addition lemma $C.1$ guarantees minimizers $g_1^*$ and $g_2^*$ such that

$$\mathrm{Obj}_{\lambda_1}^A (g_1^* \circ f_{\theta_1}) = \inf_{g \in G^S} \mathrm{Obj}_{\lambda_1}^A (g \circ f_{\theta_1}) \text{ and } \frac{1}{n} \sum_{i=1}^{n} \left\| J_{g_1^* \circ f_\theta}(x_i) \right\|_2 > 0$$

$$\mathrm{Obj}_{\lambda_2}^A (g_2^* \circ f_{\theta_1}) = \inf_{g \in G^S} \mathrm{Obj}_{\lambda_2}^A (g \circ f_{\theta_1}) \text{ and } \frac{1}{n} \sum_{i=1}^{n} \left\| J_{g_2^* \circ f_{\theta_1}}(x_i) \right\|_2 > 0$$

Thus, we have that

$$\mathrm{Obj}_{\lambda_1}^A (g_2^* \circ f_{\theta_1}) = \hat{R}(g_2^* \circ f_{\theta_1}) + \lambda_2 \cdot \epsilon \frac{1}{n} \sum_{i=1}^{n} \left\| J_{g_2^* \circ f_{\theta_1}}(x_i) \right\|_2$$

$$> \hat{R}(g_2^* \circ f_{\theta_1}) + \lambda_1 \cdot \epsilon \frac{1}{n} \sum_{i=1}^{n} \left\| J_{g_2^* \circ f_{\theta_1}}(x_i) \right\|_2 \qquad \text{since } \lambda_2 > \lambda_1$$

$$\geq \hat{R}(g_1^* \circ f_{\theta_1}) + \lambda_1 \cdot \epsilon \frac{1}{n} \sum_{i=1}^{n} \left\| J_{g_1^* \circ f_{\theta_1}}(x_i) \right\|_2 \qquad \text{def of } g_1^*$$

$$= \mathrm{Obj}_{\lambda_1}^A(\mathbf{0}) \qquad \text{lemma } C.1$$

Thus $f_{\theta_1} \notin \mathcal{F}_{\lambda_2}^A$ which implies $\mathcal{F}_{\lambda_2}^A \subsetneq \mathcal{F}_{\lambda_1}^A$. It remains to show for $\lambda_1 \geq 0$, we have that $\mathcal{F}_{\lambda_1}^A \subsetneq \mathcal{F}$.

Consider any $g \in \mathcal{G}^S$. For $j \in [L]$, define $W^j$ as the weight matrix where $W^j = I_{d \times d}$ (identity matrix) for $j \in [L-1]$ and let the final weight $W^L = B \cdot I_{d \times d}$ for some constant $B > 0$. Set the bias vectors $b^j = \mathbf{0}$ for $j \geq 2$. Let the first bias equal $b^1 = R \cdot \mathbb{1}$ where $\mathbb{1}$ is the vector of all 1's and $R$ is the upper bound such that $\|x\|_\infty \leq R$. Set $\theta = (W^1, b^1, \ldots, W^L, b^L)$ and let $h_\theta = g \circ f_\theta$

We compute

$$\mathrm{Obj}_{\lambda_1}^A(h_\theta) = \hat{R}(h_\theta) + \lambda_1 \cdot \epsilon \frac{1}{n} \sum_{i=1}^{n} \|J_{h_\theta}(x_i)\|_2$$

$$= \frac{1}{n} \sum_{i=1}^{n} \|B(x_i + R\mathbb{1}) - y_i\|^2 + \frac{1}{n} \sum_{i=1}^{n} \|J_{h_\theta}(x_i)\|_2$$

$$= \frac{1}{n} \sum_{i=1}^{n} \|B(x_i + R\mathbb{1}) - y_i\|^2 + B\|g\|_2$$

$$\geq \frac{1}{n} \sum_{i=1}^{n} \|B(x_i + R\mathbb{1}) - y_i\|^2 + B\delta$$

We note that sending $B \to \infty$ we get $\mathrm{Obj}_{\lambda_1}^A(h_\theta) \to \infty$ which implies that there exists a $B = B'$ such that $\mathrm{Obj}_{\lambda_1}^A(h_\theta) > \mathrm{Obj}_{\lambda_1}^A(\mathbf{0})$. Setting $B = B'$ implies $f_\theta \notin \mathcal{F}_{\lambda_1}^A$.

$\square$

# D   EXTRA EXPERIMENT RESULTS

## D.1   ABSOLUTE TRANSFERABILITY

We show the results with absolute transferability in Figure 7,8,9 and 10 respectively.

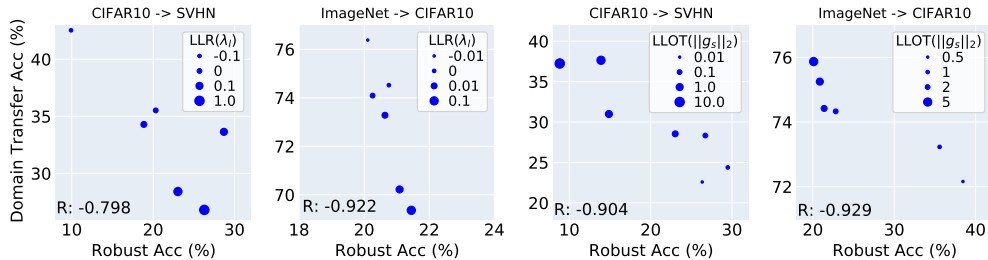

Figure 7: Robustness and absolute transferability when we control the norm of last layer with last-layer regularization (LLR) and last-layer orthogonal training (LLOT) with different parameters.

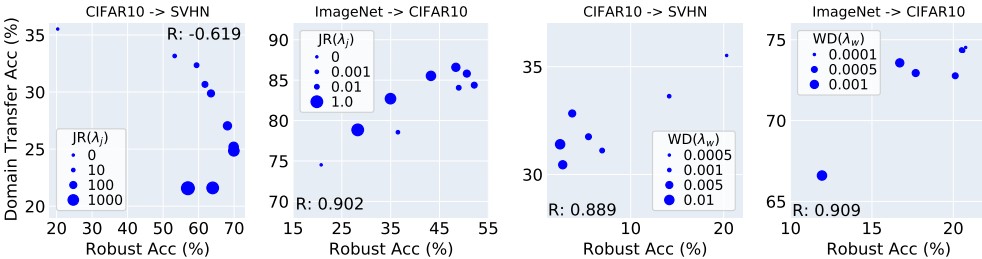

Figure 8: Robustness and absolute transferability when we regularize the feature extractor with Jacobian Regularization (JR) and weight decay (WD) with different parameters.

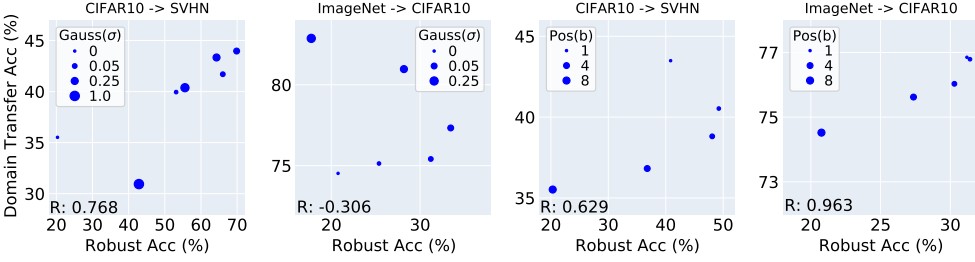

Figure 9: Robustness and absolute transferability when we use Gaussian noise (Gauss) and posterize (Pos) as data augmentations with different parameters.

## D.2   RESULTS OF OTHER MODEL STRUCTURES

To further validate our evaluation results, we evaluate the experiments on another model structure. We use a simpler CNN model for CIFAR-10 to SVHN and a more complicated WideResNet-50 for ImageNet to CIFAR-10. The CNN model consists of four convolutional layer with $3 \times 3$ kernels and 32,32,64,64 channels respectively, followed by two hidden layer with size 256. A $2 \times 2$ max pooling is calculated after the second and fourth layer. Other settings are the same as in the main text. Note that in some settings the new model cannot converge, and therefore we will omit the result. In addition, Jacobian regularization cannot be applied on WideResNet-50 because of the large memory cost, so we do not include it in the figures. The results are shown in Figure 11, 12 and 13.

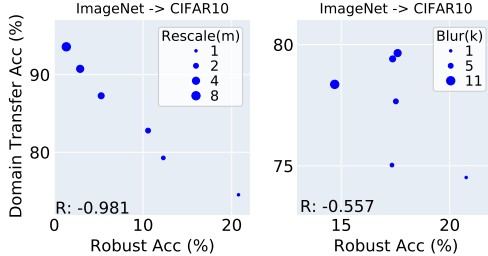

Figure 10: Robustness and absolute transferability when we use rescale and blur as data augmentations with different parameters.

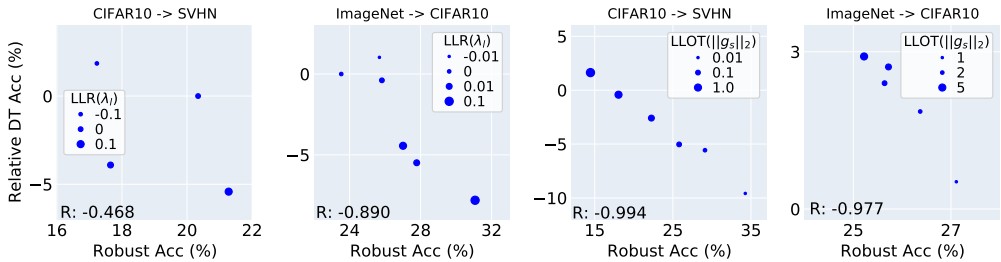

Figure 11: Robustness and transferability for the other model structure when we control the norm of last layer with last-layer regularization (LLR) and last-layer orthogonal training (LLOT) with different parameters.

## D.3 DATA AUGMENTATIONS THAT VIOLATE SUFFICIENT CONDITION

We study rotation and translation, the two data augmentations that violate the sufficient condition for regularization. The result is shown in Figure 14. We observe that these augmentations do not have an obvious impact on domain transferability.

## D.4 ROBUSTNESS EVALUATION WITH AUTOATTACK

Besides PGD attack, we also evaluate the model robustness using the stronger AutoAttack. We use APGD-CE, APGD-T and FAB-T as the sub-attacks in AutoAttack with 100 steps. Since the accuracy will decrease after the stronger attack, we use a slightly smaller $\epsilon = 0.2$ to better visualize the trend. The results are shown in Fig. 15. We can observe that the trend is similar with what we observed before when we used the PGD attack - domain generalization is an effect of regularization and data augmentation, and it is sometimes negatively correlated with model robustness. Also,

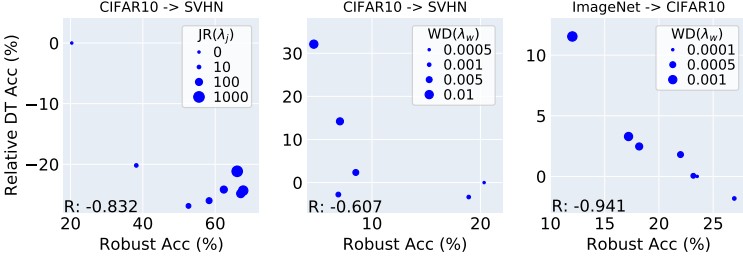

Figure 12: Robustness and transferability for the other model structure when we regularize the feature extractor with Jacobian Regularization (JR) and weight decay (WD) with different parameters.

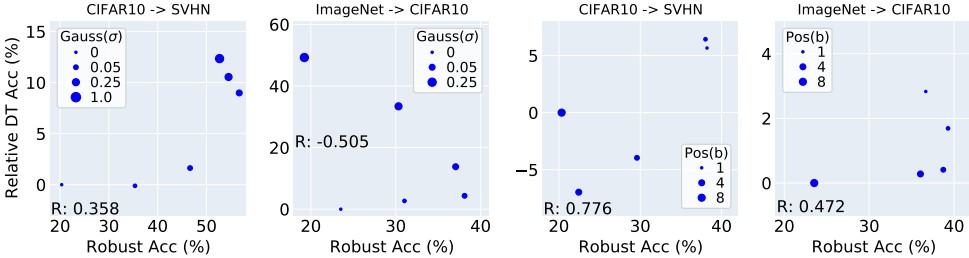

Figure 13: Robustness and transferability for the other model structure when we use Gaussian noise (Gauss) and posterize (Pos) as data augmentations with different parameters.

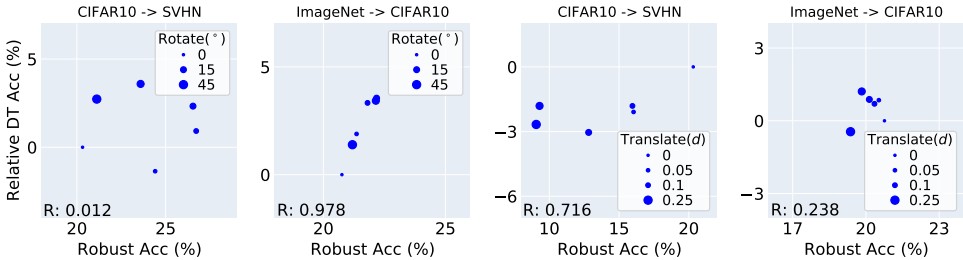

Figure 14: Relationship between robustness and transferability when we use rotation and translation as data augmentations.

augmentations like rotation and translation, which violates the sufficient condition, do not improve the domain generalization.

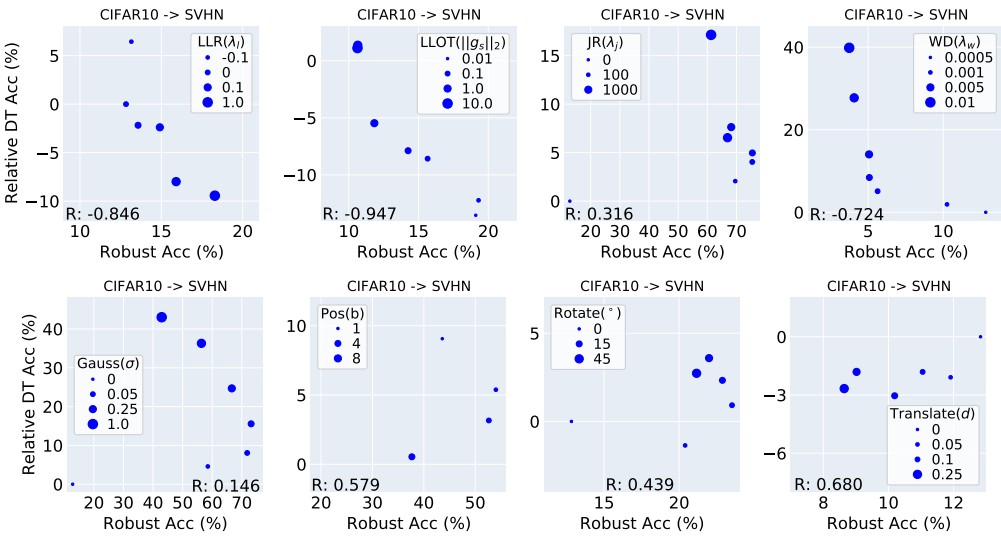

Figure 15: Relationship between robustness and transferability on CIFAR-10 when we use AutoAttack to evaluate model robustness.

