# OpenReview forum: "Adversarially Robust Models may not Transfer Better: Sufficient Conditions for Domain Transferability from the View of Regularization"
_ICLR.cc/2022/Conference — ICLR 2022 Submitted_

### Official Review · Reviewer_ijfD · 2021-10-27

**Correctness:** 3
**Technical Novelty And Significance:** 3
**Empirical Novelty And Significance:** 3
**Recommendation:** 3
**Confidence:** 3

**Main Review:**

Strengths: This paper studies the relation between robustness and generalization, which are both interesting and relevant topics in machine learning. On the theoretical side, it provides a new definition called relative domain transferability loss as well as a pseudometric, and gives generalization bound based on Rademacher complexity. On the experimental side, it says with a bigger regularization and stronger augmentation, the generalization measured by relative DT accuracy improves.

Weaknesses:

This paper is not clearly written and there are many confusing parts. See below:
1. Clarity: I'm not sure how the robustness is defined in this paper. Throughout the paper, I only see the discussion on the regularization (either Jacobian or last-layer) and the data augmentation, but how they correlate with robustness is unclear. Is the robustness mentioned distributional robustness or robustness wrt sample perturbation? More clarification is needed.
2. Clarity: In Sec 3.1, are you doing regression or classification? In Sec 4 the experiments are classification but here you are treating the label $y_S$ as real number. Shouldn't the predictor $f$ output a label?
3. Clarity: Prop 3.1, what is the relative domain transferability? Any reference?
4. Clarity: I'm not sure about the motivation of Def 2. In domain adaptation there has been many papers that study the distance metric, e.g.:

[1] Ben-David et al, A theory of learning from different domains, Machine Learning 2010.

[2] Mansour, Y., Mohri, M., and Rostamizadeh, A. Domain adaptation: Learning bounds and algorithms. COLT 2009.

[3] D. Acuna, G. Zhang, M. Law and S. Fidler, f-Domain-Adversarial Learning: Theory and Algorithms, ICML 2021.

Could you compare your Def 2 with existing metrics?

5. Clarity: In Theorem 3.2, what is individual loss function? How would it relate to $\ell_{D_S}$?
6. Minors: Def 3: reference for Rademacher complexity is needed; Prop 3.2, $D'$ is not defined;
7. Clarity: in the experimental section, it is observed that wth stronger regularization/augmentation the DT acc is higher. However, the meanings of Relative DT Acc and Robust Acc are not clear or not well-explained in the paper.
8. Clarity: how would your theory help in the experiments? The connection between regularization and the proposed pseudo metric is not clear.
9. Novelty: how would you compare the conclusion in your paper with Salman et al 2020? In their paper it is claimed that robustness helps generalization, which seems to be contradictory.


**Summary Of The Paper:**

This paper aims to convey the message that adversarially robust models may not have better transferability in terms of transfer learning in vision tasks. However, this message is unclear due to the mixture of data augmentation, regularization, robustness in the presentation. Some definitions are confusing and the conclusion seems to contradict with existing literature, e.g,  Salman et al arxiv: 2007.08489.

--post rebuttal--

Thanks to the authors for the draft revision and the additional appendix C. Now the paper is clearer to me: the general idea is to say that robustness is a type of regularization, and this regularization is the key to generalization. However, I think the former is not novel as already mentioned in Roth et al NeurIPS 2020. Besides, there are still many presentation issues even after the revision, e.g.:
1) Prop 3.1 mentioned the relative domain transferability, but the formal definition is introduced later in Def 1.
2) The added paragraph is confusing to me. How would you define the input space ${\cal X}$? Is it a compact set? Why would the perturbation go outside ${\cal X}$? Shouldn't $x + \delta$ still be in the domain of $f_c^{{\cal D}_{\cal S}}$ as otherwise it is not well-defined?
3) Def 2 should be compared with existing metrics.
4) In Section 3.4 the authors use the squared loss but in experiments they use cross entropy loss.

Despite the interesting topic, I would keep my current score until a more well-written version is presented.



**Summary Of The Review:**

Overall this paper aims to address an important issue: does robustness help generalization? However, in practice it mostly does experiments on regularization and data augmentation, which may not be directly related to robustness. Comparison with existing work is needed, and the presentation (especially some introduction of Theorems and Definitions) needs to be improved. Therefore I would not recommend acceptance.

---

> ### Author Response · Authors · 2021-11-20
> **Response to Reviewer ijfD**
>
> We thank the reviewer for the valuable feedback. The revisions of the paper are highlighted in blue. Our response to the major comments is as follows:
>
> ** 1. Definition of Robustness ** (I'm not sure how the robustness is defined in this paper.)
>
> There are two places where we introduce the robustness. As for the example in section 3.1, the robustness is the adversarial robustness against input perturbations. We have made the definition more clear in the revised paper in section 3.1. As for the experiments, we follow the setting in [1].
>
>
> ** 2. In Sec 3.1, are you doing regression or classification? **
>
> Thanks for pointing this out. In section 3.1, the example setting is quite general: finding a model (function) in a function space that minimizes a norm distance. Therefore, it applies to either classification (the ground-truth function $y$ being discrete) or regression (the ground-truth function $y$ being continuous), although the main paper mainly focuses on classification models and we have made this clear in our revision.
>
>
>
> ** 3. Prop 3.1, what is the relative domain transferability? Any reference? **
>
> The relative domain transferability loss measures the difference between the loss of the source model on the target domain and the loss of the source model on the source domain. A formal definition of it is Definition 1 in our paper. We show that the relationship of regularization to the relative domain transferability is more reasonable to analyze since after applying the regularization both the source model's loss on the target and source domains could change. We provided the related discussions in the interpretation of Theorem 3.1.
>
>
> ** 4. Comparison Def 2 with existing metrics ** (Motivation of the (G, F)-pseudometric and its comparison with existing metrics)
>
> The motivation of the (G, F)-pseudometric comes from the following observations. We want to study what factors affect how a source model transfers to the target domain. The obvious factor is the difference between the two domains. But the function class where the model is trained from is also an important factor (e.g., see the example in section 3.1). To consider both of the factors, we propose the (G, F)-pseudometric.
>
> The major difference with existing metrics is that the proposed (G, F)-pseudometric is more general than the existing metrics by Ben-David et al., Mansour et al. and  Acuna et al. as the reviewer cited. Concretely, the three works only consider the distributions on the input space $\mathcal{X}$, while we consider the distributions on both the input space and the output space, i.e, $\mathcal{X}\times\mathcal{Y}$. This difference enables us to consider the fine-tuning process, which is important and widely applied in practice.
>
>
>
> ** 5. In Theorem 3.2, what is the individual loss function? How would it relate to $\ell_{\mathcal{D}_S}$? **
>
> The individual loss function $\ell:\mathcal{Y}\times\mathcal{Y}\to R_+$ is the loss function that acts on a prediction and its ground-truth target. The relation between $\ell$ and $\ell_{\mathcal{D}_S}$ is defined in the notation paragraph of section 3 (the second paragraph of section 3).
>
> ** 6. Minor: reference for Rademacher complexity is needed; Prop 3.2, D′ is not defined;**
>
> Thanks for pointing it out. We have included it in the revised version.
>
> ** 7. The meaning of relative DT acc and robust acc**
>
> We introduce how we evaluate relative DT acc and robust acc in Sec. 4.1. The relative DT acc of one model is defined as: $(acc_{tgt} - acc_{src}) - (acc_{tgt}^v - acc_{src}^v)$, where the $acc^v$ is the accuracy of vanilla model. The robust accuracy is evaluated as the model accuracy under PGD attack configured in the paper, following the standard setting as in [2].
> [2] Salman H, Ilyas A, Engstrom L, et al. Do adversarially robust imagenet models transfer better? arXiv preprint arXiv:2007.08489, 2020.
>
> ** 8. Relationship between theory and experiments ** (how would your theory help in the experiments?)
>
> Thanks for the insightful question. Our experiment setting is fully motivated by our theory. In the theory, we find that model regularization and data augmentation will help with its relative domain transferability. Thus, we train the models with different regularization and augmentation and thus evaluate the relationship between their transferability and robustness which verifies our theory. As for the metrics, we follow the standard-setting as in [1] to show the trend of domain transferability and model robustness.

---

> > ### Author Response · Authors · 2021-11-20
> > **Response to Reviewer ijfD (2/2)**
> >
> > ** 9. Comparison with Salman et al 2020 ** (In their paper it is claimed that robustness helps generalization, which seems to be contradictory.)
> >
> > Thanks for the helpful comment and yes the reviewer is exactly correct on the conclusion. Our paper is motivated by and proposed as a counter-example for Salman et al 2020. In their paper, they claim that robust models will transfer better. However, we aim to show in our paper that robustness is not the causal factor of domain transferability -- the improved domain transferability is the result of adversarial training as a regularization (Please see the new results in appendix C).
> >
> > Overall, we show both theoretically and empirically that the function class of the feature extractor, which can be realized by different regularizations and data augmentations, is the sufficient condition to domain transferability, rather than model robustness. In addition, we have conducted extensive experiments under different settings to show such counter-examples (i.e. more robust models are less domain transferable) based on Salman et al 2020, which are supported by our theory. In the revised version, we emphasize this motivation in the introduction and mention that our paper provides counter-examples for Salman et al 2020.
> >
> > [1] Ilyas et al. Adversarial examples are not bugs, they are features.

---

### Official Review · Reviewer_nSDW · 2021-11-02

**Correctness:** 3
**Technical Novelty And Significance:** 3
**Empirical Novelty And Significance:** 2
**Recommendation:** 5
**Confidence:** 3

**Main Review:**

This paper investigates a very interesting problem from a theoretical standpoint. Author(s) have been somehow successful in bounding the domain generalization loss from a source domain to a target, via a number of newly-defined (and possibly fundamental) measures which involve the distributions and function sets of the problem configuration. In particular, a new (as far as I am aware) measure has been proposed which mimics the role of Rademacher complexity, but this time works when one wants to transfer a learned model from a domain S to a target domain T. I haven't completely checked the proofs yet, but results seem legit.


The main problem is that although the overall theoretical framework is interesting, but results might not be that much strong at the moment With some extra time and effort, far more interesting achievements can be reached. For example, the $\left(\mathcal{G},\mathcal{F}\right)$-pseudometric between source and domain distributions, i.e., $\mathcal{D}_S$ and $\mathcal{D}_T$, can be more mathematically analyzed and perhaps new theoretical insights might become available. In particular, how the similarity (for example in terms of TV distance) between $\mathcal{D}_S$ and $\mathcal{D}_T$ affects this measure? In another example, showing that adding more regularization on the feature extractor gives better "domain generalization" is not enough; Since this might degrade the overall performance in return. Please note that the constant function $f(x)=0$ performs equally bad on all domains. However, author(s) have not discussed this issue in details.

The other problem is that there are numerous grammatical errors and also a number of minor flaws inside the main body. Writing of the paper should be completely reviewed. I have pointed to some of the grammatical errors below in the "Minor comments" section. Also, some minor technical ambiguities are noticeable: At some point, authors have referred to Vapnik-Chervonenkis (VC) dimension of a function set, which includes functions from $\mathcal{X}$ to $\mathbb{R}^{+}$. Based on my understanding of SLT, VC-dimension can only be defined for a class of binary classifiers, and not functions that output continuous values.

-----------------------------------------------------------------------------------------------------------------------------------

Minor comments and suggestions:
- Abstract -> "On one hand,"
- Abstract -> "Norm of last layer norm"?
- Introduction -> please refrain from underlining long phrases for the sake of emphasis. Instead, you can use quotation marks ``...", or the italic form.
- Section 2 (Related Works) has been poorly organized. Sentences are vague, unrelated to one another, and need much more discussion to convey meaningful information to the reader. Fo example, what exactly "generalizing beyond convex hull" refers to here? Or, what are $\mathcal{H}$-divergences.
- Please use \ eqref{} (and not \ ref{}) to refer to equations. For example, your first reference to equation (1) is not standard.

**Summary Of The Paper:**

This paper aims to investigate the theoretical connection between domain generalization (aka Transferability) and adversarial robustness in some general settings. Authors claim that a thorough theoretical treatment of this problem has not been given yet, and therefore set out to establish a number of fundamental relations.

A simple example has been proposed which shows the existence of cases, where adversarial robustness and transferability can be independent (or even negatively correlated). Also, paper proves that adding more restriction (tighter regularization) on the feature extractor stage of a learning algorithm gives better domain generalization. Additionally, some intrinsic and fundamental measures have been defined to bound the domain generalization error for transferring a learned model from domain S to domain T. In this regard, uniform convergence bounds have been derived to show the gap between empirical and statistical versions of such measures remain small or even converge to zero when sample size asymptotically increases.

Finally, a number of experimental results have been shown to support the above theoretical achievements. I have not gone through the experimental parts nor the proofs, yet.

**Summary Of The Review:**

The proposed analysis, in its current shape and form, might not be ready for publication at ICLR. While the proposed framework in this paper is interesting and definitely worthy of more analysis, the current results including all high-probability concentration bounds and etc. are still incremental and need more work.

The other problem is writing. There are grammatical errors throughout the paper and the quality of writings can be greatly improved.

Overall, I believe the paper is not ready for publication yet. My vote is weak reject.

---

> ### Author Response · Authors · 2021-11-20
> **Response to Reviewer nSDW**
>
> We thank the reviewer for the valuable and constructive feedback. The revisions of the paper are highlighted in blue. Our response to the major comments is as follows:
>
> ** 1. Further exploration ** (the (G,F)-pseudometric between source and domain distributions, i.e., DS and DT, can be more mathematically analyzed and perhaps new theoretical insights might become available. In particular, how the similarity (for example in terms of TV distance) between DS and DT affects this measure?)
>
> We thank the reviewer for the thoughtful suggestion. The proposed (G, F)-pseudometric is both a complexity measure of the model function class and a distance measure of two distributions. Given a certain fixed function class, (G, F)-pseudometric can serve as a distance measure like the TV distance.  After further investigation, we find that the (G, F)-pseudometric is closely related to the Wasserstein distance between the two domain distributions $\mathcal{D}_S, \mathcal{D}_T$ and also the total variation distance. Two new propositions are added in the revised paper:
>
> In proposition A.3 we show that the (G, F)-pseudometric between $\mathcal{D}_S$ and $\mathcal{D}_T$ is upper bounded by $LW(\mathcal{D}_S, \mathcal{D}_T)$, assuming the loss function class is $L$-Lipschitz. It is known that the Wasserstein distance is closely related to the total variation (TV) between the two distributions. For example, Wasserstein distance can be upper bounded by the TV under some mild conditions (Case 6.16 in Optimal Transport Old and New by Villani).
>
> In proposition A.4, if we are working in the realm of multi-class classification and the loss function is the 0-1 loss, we show that the total variation distance upper bounds the (G, F)-pseudometric.
>
> ** 2. More regularization might degrade the overall performance ** (Adding more regularization on the feature extractor gives better "domain generalization" is not enough; Since this might degrade the overall performance in return.)
>
> We would like to clarify that we do not claim a universal monotonic relationship between the regularization and performance. Instead, our theory and experiments suggest a monotonic relationship between the regularization and relative performance under common ranges. We are aware that too much regularization may hurt the absolute performance, and the performance criterion in our experiments is the relative domain transferability. We provied the discussion about the trade-off between the regularization and the absolute performance in the interpretation of Theorem 3, and we have emphasized this point in the revised version.

---

### Official Review · Reviewer_MEJz · 2021-11-02

**Correctness:** 2
**Technical Novelty And Significance:** 2
**Empirical Novelty And Significance:** 3
**Recommendation:** 3
**Confidence:** 4

**Main Review:**

The paper is well-written.

However, there are several big problems:

1. Overselling: transfer better due to regularization is not initiated by this work as they claim, for instance, Adversarial Training Helps Transfer Learning via Better Representations by Deng et al. have theoretically analyzed the regularization effect of adversarial training. Also, data augmentation served as regularization has also been extensively studied. Such as in On Mixup Regularization by Carratino, L et.al, Dropout Training as Adaptive Regularization by Wager et al.
More discussion is needed
2. The counterexample in 3.1, the explanation and intuition is not right, delta wishes to maximize the loss, instead, the author say there exists delta such that example is pushed outside the manifold of input, so the robustness could be arbitrarily strong. This intuition is completely mysterious.
3. The monotonic relationship between regularization and performance is not quite right, since it is common sense that too strong regularization could result in bad performance. Thus, this claim is a little too exaggerated.
4. The tight upper bound has no lower bound to support. The tightness is not rigorously justified.



**Summary Of The Paper:**

This paper studies theoretically how adversarially trained models can transfer better, and disentangle the robustness and accuracy on the target domain. They claim that the main reason is more of regularisation rather than robustness. Corresponding examples and theory are presented.

**Summary Of The Review:**

This paper is overclaiming their contribution. Details are in the main review.

---

> ### Author Response · Authors · 2021-11-20
> **Response to Reviewer MEJz**
>
> We thank the reviewer for the valuable feedback. The revisions of the paper are highlighted in blue. Our response is as follows:
>
> ** 1. Prior work ** (More discussion about the papers by Deng et al., Carratino et al. and Wager et al.)
>
> Thank you for the helpful comment. We believe that our paper addresses a unique perspective different from these papers, and we will cite the papers and add discussion in our related work. Concretely, Deng et al. only analyze how adv training helps with domain transfer, not in the view of regularization. Carratino et al. analyze the Mixup and some other data augmentations, but not the relationship with domain transfer. The paper by Wager et al. analyzes the dropout as a regularization, but not related with domain transfer. Moreover, the regularization effect that Carratino et al. or Wager et al. considers does not focus on the feature extractor which is the key to domain transferability. In addition, the two papers respectively analyze Mixup and Dropout, which are very specific DA methods, but our analysis of data augmentation provides a general understanding on sufficient conditions for the family of linear augmentations.
>
>
> ** 2. Intuition of the counterexample in section 3.1 ** (Delta wishes to maximize the loss, instead, the author says there exists delta such that the example is pushed outside the manifold of input, so the robustness could be arbitrarily strong.)
>
> The intuition and goal of this example is to show that domain transferability and robustness are independent, (e.g., the first sentence of section 3.1). Since the objective only forces the learner to find a source model that performs well only on the data manifold, the learned model can be arbitrarily bad outside the data manifold. Therefore, for learned models whose loss outside the data manifold is larger than the loss inside the manifold, the delta that maximizes loss pushes the example outside the manifold. That is to say, the example shows that robustness could be arbitrarily bad without affecting the domain transferability, illustrating that the two properties can be independent of each other. We have updated our paper to make this discussion clear.
>
>
> ** 3. Relationship between regularization and performance ** (The monotonic relationship between regularization and performance is not quite right, since it is common sense that too strong regularization could result in bad performance.)
>
> We would like to clarify that we do not claim a monotonic relationship between the regularization and performance. Instead, our theory suggests a monotonic relationship between the regularization and relative performance under common conditions as we verified in the experiments. We are aware that too much regularization may hurt the absolute performance, and the performance criterion in our experiments is the relative domain transferability. In fact, we provided discussion about the trade-off between the regularization and the absolute performance in the interpretation of Theorem 3. We have emphasized this point in the revised version.
>
>
> ** 4. Lower bound of the tight upper bound ** (The tight upper bound has no lower bound to support. The tightness is not rigorously justified.)
>
> We note that Theorem 3.2 implies a lower bound. The equivalent statement of Theorem 3.2 is: under the same condition , the relative domain transferability $\tau(A; D_S, D_T)$ is both upper and lower bounded:
> $$d_{F_A}(D_S, D_T)-\epsilon\leq  \tau(A; D_S, D_T) \leq d_{F_A}(D_S, D_T),$$
> where $\epsilon$ can be arbitrary close to $0$. It thus justifies the tightness of the upper bound $d_{F_A}(D_S, D_T)$. We have made it more clear in the revised paper.

---

### Official Review · Reviewer_5SD9 · 2021-11-03

**Correctness:** 3
**Technical Novelty And Significance:** 3
**Empirical Novelty And Significance:** 3
**Recommendation:** 6
**Confidence:** 4

**Main Review:**

Strengths:
1. This paper challenges a fundamental claim in ML: stronger robustness leads to better domain generalization, and gives a solid analysis.
2. This paper presents a theoretical framework to analyze the relationship between regularization strength and domain transferability.
3. The empirical experiments are also very interesting. Specifically, they show that the robustness does not necessarily correlate to domain transferability.

Weaknesses/Question:
1. How to obtain robust accuracy? Using PGD to evaluate the robust accuracy while controlling the norm of the last layer is very tricky. See the temperature scaling attack here https://arxiv.org/pdf/1607.04311 . This is my biggest concern, please validate the results using a more advanced attack method, e.g., AutoAttack. If you can obtain the same conclusion with AutoAttack, I will raise my rating.






**Summary Of The Paper:**

This paper discusses a very interesting question: what is the relationship between adversarial robustness and cross-domain transferability. The previous studies show that a more robust model can transfer better. This paper argues that the true reason for the cross-domain transferability is not the adversarial robustness, but the effect of regularization, which can also be achieved through other methods, such as data augmentation.

**Summary Of The Review:**

This paper provides a great theoretical framework for analyzing domain transferability and explains why improving adversarial robustness can improve domain generalization. The empirical experiments are very interesting. However, the method of evaluating the adversarial robustness is not very appropriate. This is very important, as it may alter the conclusion of this paper, especially for the norm controlling experiments. If the author can obtain the same conclusion with a more advanced attack method (e.g., AutoAttack), I will raise my rating.

---

> ### Author Response · Authors · 2021-11-20
> **Response to Reviewer 5SD9**
>
> We thank the reviewer for the valuable feedback. The revisions of the paper following the suggestion are highlighted in blue. Our response is as follows:
>
> ** Evaluation with more advanced attack method ** (please validate the results using a more advanced attack method, e.g., AutoAttack.)
>
> Thanks for the suggestion! We updated our paper and added the evaluation with AutoAttack in Figure 15 in Appendix D.4. From the results, we can observe that the conclusions are the same -- domain generalization is an effect of regularization and data augmentation, and adversarially robust models may not always transfer better.

---

> > ### Comment · Reviewer_5SD9 · 2021-11-28
> > **Thanks for your response**
> >
> > The additional results looks good. I have no more major concerns.
> > But other reviewers seem have many comments. Please discuss with other reviewers.
> > I set my rating as 6.

---

### Decision · Program_Chairs · 2022-01-20

**Decision:**

Reject

**Comment:**

The paper tries to analyze the relationship between regularization, adversarial robustness, and transferability.

Pros:
- An interesting problem was tackled.

Cons:
- The main claim (Prop.3.1) is almost trivial.  Prop. 3.1 shows that "relative" transferability is smaller for stronger regulariation, which is just a slight generalization of the triangler inequality ||YT = YS|| <= ||YT - Y|| + ||YS - Y|| for any Y in Fig.2.
- Experiments show negative correaltion between the relative transferability and accuracy, which is trivial.  Large regularization degrades the accuracy which increases the "relative transferability".  "Absolute" transferability in Appendix doesn't show clear negative correlations.
- Salmann et al. claimed that adversarially "trained" models transfer better, and did not claim that there are positive correlations between the transferability and robustness for general classifiers without adversarial training.  So the finding in this paper is not surprising nor against Salmann et al.

To prove that adversarial robustness is just a subproduct of regularization, the authors should show that the "absolute" transferability by adversarially trained classifier can be achieved by other regularization.  Defining relative transferability is fine if it is just a decomposition to conduct an analysis of the absolute transferability.  But no conclusion on the performance should be made from its analysis, because a trivial correlation will appear, i.e., (A-B) and B should be negatively correlated unless A strongly correlates to B.  Also, this is highly misleading so that some reviewers seem to have misunderstood that the authors would have claimed that negative correlations between regularization and absolute transferability were observed in the original submission.

Overall, the paper requires major revision.